# Concurrent warming, freshening and cessation of deep convection in the Labrador Sea raised its sea level to a record high

Igor Yashayaev [1] ✉ & Yang Zhang [2,3]

The Labrador Sea, a major North Atlantic carbon sink and source of ventilated intermediate-depth water masses, is a vital component of the global climate system. Since the 1950s, it has seen significant heat and freshwater content shifts, resulting in arguably the largest full-depth oceanic temperature and salinity changes ever recorded. Here, we quantitatively assess the relative contributions of these changes to sea level variability. Using satellite altimetry in conjunction with profiling Argo float and ship-based hydrographic measurements, we show that between 2017 and 2025 the central Labrador Sea experienced an exceptionally fast sea level rise to record high. Six concurrent factors contributed to this – reduced winter cooling, enhanced summer warming, anomalous freshening, ceased deep convection, reduced deep-water density, and water-column mass gain. The temperature-driven sea level changes are controlled by surface heat fluxes. The salinity effects switched from counterbalancing temperature effect (1948–2015) to reinforcing (2015–2023), making the unprecedented Labrador Sea freshening and feeding it extreme Arctic sea ice losses (with a two-year lag) essential contributors to the 2017–2025 sea level rise.

The Earth's climate system is in a state of imbalance, with excess heat accumulating at a rate of 0.5–1 W/m² since the 1970s, approximately 90% of which has been absorbed by the ocean[1–7]. The excessive heat accumulation has not only contributed to ocean warming but also accelerated the melting of glaciers, ice sheets, and sea ice, and the hydrologic cycle of the Earth[8], significantly impacting oceanic freshwater content, particularly in high-latitude regions[9–12]. Sea-level rise is one of the most significant consequences of these trends, resulting from the thermal expansion of seawater caused by ocean warming and the addition of freshwater from the melting land-ice[13–15]. Additionally, the planetary distribution of freshwater can be shifted by the accelerating hydrologic cycle[8]. Regionally, freshening caused by melting sea ice also contributes to seawater expansion and sea-level rise. The Labrador Sea, which has the lowest mean dynamic sea level in the Northern Hemisphere, is a unique region where sea level is influenced by a combination of these factors.

The Labrador Sea is located in the western part of the subpolar North Atlantic and bounded by the cold fresh Arctic waters carried by the West Greenland Current (WGC) to the northeast and Labrador Current (LC) to the southwest (Fig. 1a). This semi-enclosed sea acts as a primary and largest *receiving basin* of the North Atlantic for meltwater produced by melting and retreating Arctic sea ice[16,17], enhanced by Arctic amplification[18], and Greenland Ice Sheet[19,20]. In general, the Labrador Sea receives the inflows of modified yet still relatively warm and saline Atlantic Water, shallow outflows and deep overflows of cold and fresh Arctic water, and continental freshwater runoffs from rivers, groundwater, and glacier and permafrost meltwater[19,21–23]. The part of the sea, defined as the central Labrador Sea (CLS, red contour in Fig. 1), is also known for its other vital function—intermediate and deep-water

[1]Bedford Institute of Oceanography, Fisheries and Oceans Canada, Dartmouth, NS, Canada. [2]School of Marine Science and Policy, University of Delaware, Lewes, Newark, DE, USA. [3]Scripps Institution of Oceanography, University of California, San Diego, La Jolla, CA, USA. ✉e-mail: Igor.Yashayaev@dfo-mpo.gc.ca

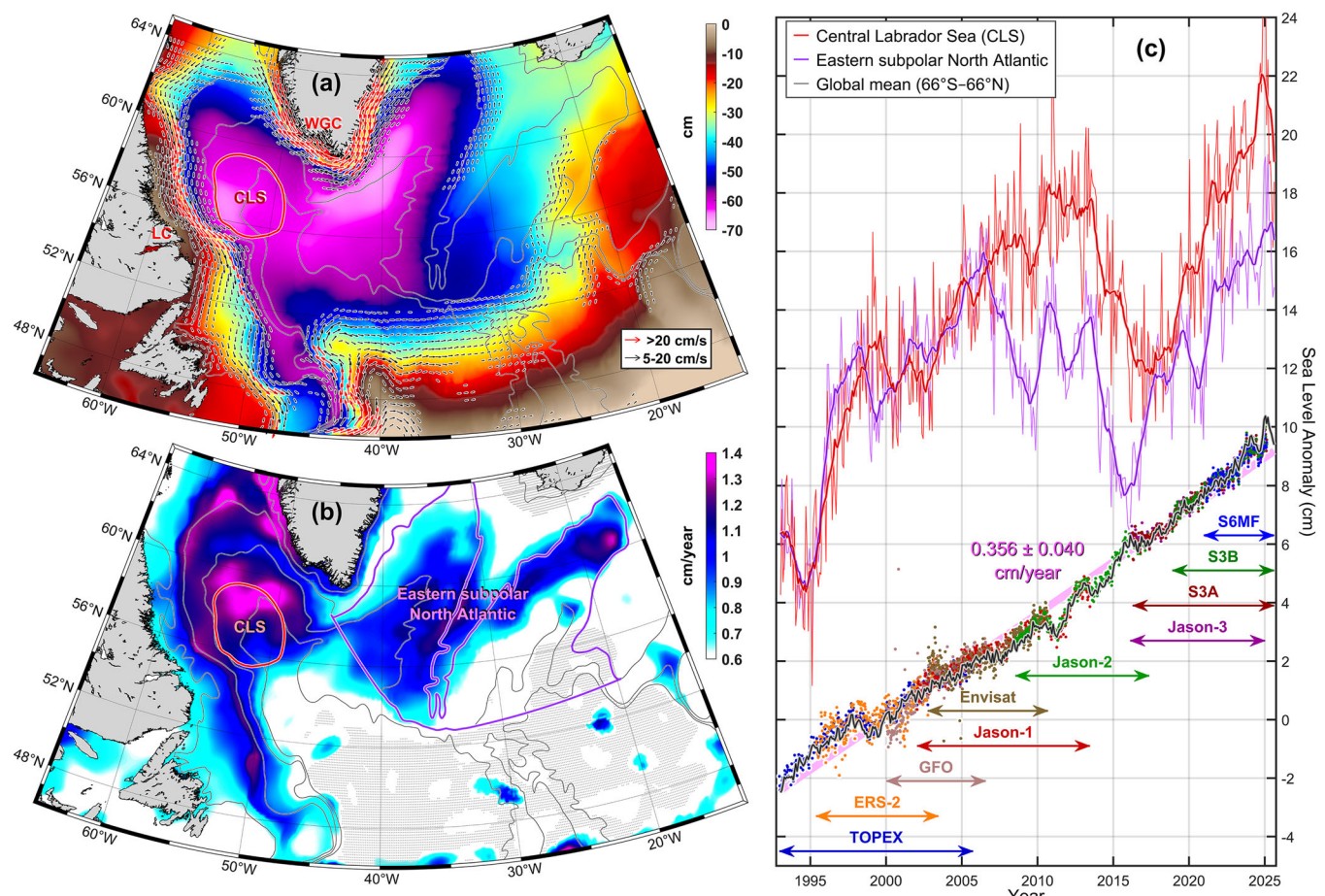

**Fig. 1 | Topographic features, surface geostrophic currents, and interannual sea level variability in the subpolar North Atlantic, and global mean sea level rise.** The 1993–2025 mean annual sea level low, major currents (*arrows*), and 200, 2000, 3000, and 3500 m bathymetric contours (*black lines*) (**a**); 2017–2025 linear regression-based sea level rate change (*gray dots indicate insignificant values*) (**b**); 1993–2025 gridded altimetry-based central Labrador Sea monthly and yearly (*red lines*), eastern subpolar North Atlantic monthly and yearly (*purple lines*), and +0.3 mm/year GIA-corrected global monthly (*black line*) mean sea level (**c**). The global mean values are cross-validated with multiple mission data provided by the NOAA Laboratory for Satellite Altimetry (**c**, *scatter dots, with major altimetry missions and corresponding time span labeled*).

mass production. This is where the waters with different origins, from tropical to polar, transform and blend, fusing together into a new water mass—Labrador Sea Water (LSW)[24].

The CLS is the primary location of the North Atlantic where intense winter cooling of the sea surface through air-sea exchange leads to vigorous convective mixing and deep-water ventilation, hence regarding it as the ocean's *lung*. The strong and cold northerly-to-westerly winds, prevailing through a typical Labrador Sea winter, extract enormous amounts of heat from the surface. This surface heat loss makes the surface water cool, densify, and sink while mixing and blending with the underlying waters of diverse origin and properties. Each winter, this mixing process—deep convection—forms a 500-to-2500 m thick thoroughly homogenized (well-mixed) layer—LSW[24]. On the other hand, the same mixing process transfers large quantities of heat accumulated between winters in a large part of the water column up to the sea surface, consequently releasing it to the atmosphere[21,24]. Furthermore, the CLS, with its sea level, i.e., dynamic topography, forming a regional low (on account of the same strong winter cooling), controls the boundary currents and, therefore, larger-scale circulation, functioning as the ocean's *heart*.

LSW is a prominent, dense, relatively fresh, highly oxygenated intermediate water mass of the North Atlantic. The properties and volume of each new LSW vintage are shaped by local atmospheric cooling, advection of cold fresh Arctic and warm saline Atlantic waters, and the water column's retention of low stratification from

previous convective events—a process known as convective preconditioning[21,24–27]. LSW, carrying its distinct physical, chemical, and biological signatures, replenishes the lower limb of the Atlantic Meridional Overturning Circulation (AMOC)[21,28], signifying the role of the CLS in the planetary ocean and climate systems. Unless specified otherwise, the values and statements in this study pertain to the CLS.

Although the formation, variability, and spreading of the LSW have been studied in detail[21,24–27,29], sea level—one of the key climate variables—has not been investigated for the CLS since 2006[30]. Among many important implications, the CLS sea level changes (a regional climatological low in Fig. 1a) influence the shoreward sea level gradients (Fig. 1a, b), which in turn impact the transport of volume, freshwater, and heat by the geostrophic boundary currents. Notably, the sea level rise rate in the CLS outpaces that of other deep subpolar North Atlantic basins (i.e., Irminger Sea and Iceland Basin) in both the long-term record[31,32] and more prominently since 2017 (Fig. 1b, c). The post-2017 CLS sea level rise rate that reaches 1.4 cm/year (Fig. 1b) is almost four times the global mean rise rate of 0.36 ± 0.04 cm/year (Fig. 1c), and falls well above the global trend range of 0.20 to 0.45 cm/year estimate for shorter time periods[33]. While strong interannual variability in the broader subpolar North Atlantic (SPNA) may reduce pentadal (e.g., 2011–2016), and even decadal (e.g., 2007–2017), changes to strongly negative, the cumulative sea level rise since 1993 remains higher in both the CLS (~18 cm) and SPNA East (~13 cm) compared to the global mean (~12 cm). As a uniquely sensitive basin,

the CLS exhibits regional sea level dynamics that differ markedly from broader basin-scale or global patterns, which calls for a focused examination of the local sea level budget and its underlying drivers.

Two factors drive sea level change: density-induced steric height variation and mass changes[14,15]. Total steric height or surface-to-bottom integrated specific volume (i.e., inverse density, Eqs. 1–3 in "*Methods*"), that is hypothesized to dominate dynamic sea level variability in the Labrador Sea, is influenced by both air-sea exchange and horizontal advection of heat, freshwater and salt carried by the ocean currents[14,34–37], with heat and freshwater fluxes directly impacting temperature, salinity, and thus full-depth steric height. The temperature-based thermosteric and salinity-based halosteric components were earlier shown to generally counterbalance each other in the subpolar North Atlantic[37–40], making the sea level rise not as fast as it would be without the halosteric compensation of the thermosteric drive. However, observational evidence, presented and discussed in this paper, implies that the halosteric component has recently switched from moderating to reinforcing the sea level rise.

Key research questions listed below and addressed in the "Results" section follow from an examination of the 2002–2025 CLS hydrographic changes.

The CLS temperature, salinity and density, hereafter, hydrographic variables (e.g., Fig. 2 and Supplementary Figs. 1–5) are governed by heat, salt and freshwater fluxes, which are in turn driven by local and remote air-sea-land-ice exchanges, ocean currents, and vertical and horizontal mixing. Year-round profiling Argo and episodic ship-based measurements allow us to resolve hydrographic changes on the weekly-to-biweekly timescales as those shown in Fig. 2. While the pre-2024 content of this figure has already been detailly discussed[24,27,41], here we extend it to September 2025, and highlight the two recent hydrographic developments that significantly changed the vertical stratification and raised the sea level to a record high: shutdown of winter convection (2021 and 2023–2025), and sustained freshening of the 0–800 m layer (2023–2025).

Intense CLS winter surface cooling induces convective mixing, which homogenizes and cools the upper and intermediate layers, whereas salinity and density are also affected by the freshwater flux, making the patterns of interannual changes different in the variables shown in Fig. 2. On the other hand, all hydrographic variables undergo similar transformations during winter, implying that: convection penetrates deeper than 400 m in all winters (short horizontal bars in Fig. 2 and Supplementary Fig. 1); the shallowest (500 m) and deepest (2500 m) winter convections since 1948 occurred in 2025 (Fig. 2 and Supplementary Figs. 1-4) and 1994[24] (Supplementary Figs. 1 and 2), respectively; while the deepest (2000 m) convection since 1995 occurred in 2018. The development, strength, and depth of each annual convective mixing event are jointly (positively or negatively) controlled by the strength of *present-year winter surface cooling*; the extent of *preconditioning of the water column by convections of the previous years*; and the amount of *buoyancy gained by the upper layer* through warming and freshening over the preceding summer-fall period, known as *restratification*[24]. Remarkably, there are cyclic multiyear developments of winter convection (Fig. 2 and Supplementary Figs. 1 and 2), known as convective cycles[24]. Wholly mapped with a consistently high temporal resolution (Fig. 2), the 2012–2025 convective cycle consists of the phases of recurrently intensifying and deepening convective mixing, from 2012 to 2018, and convective relaxation, from 2019 to 2025, culminating with full shutdowns of deep, *i.e., deeper than 1000 m*, convection in 2021 and 2023–2025 (Supplementary Fig. 3).

The essence of this water-mass development cycle is captured in the *bottom panel* of Fig. 2, and Supplementary Fig. 5, illustrating the evolution of equisized density layer thicknesses within and across years. The density and thickness of the recurrently mixed homogenized thick layer, termed as the pycnostad, are key to sea level

analyses. Stronger and deeper convection leads to a denser, thicker pycnostad, while weaker mixing produces a thinner, lighter one. In turn, a denser and thicker pycnostad increases the average density over the 0–2000 dbar layer, and vice versa. The pycnostad's multiyear evolution shows how progressively deepening, densifying convective vintages constituted a voluminous LSW class, and how it all collapsed afterwards, ceasing the production of thick dense LSW, especially in 2025. Notably, it was predicted in ref. 24 that deep convection would remain shut beyond 2023. The updated CLS time series shows no indication of deep convection in both 2024 and 2025, and hence confirms this prediction.

The upper 800 m layer of the Labrador Sea underwent progressive freshening from 2018 to 2023, reaching a record low salinity in 2023[24], which persisted through 2025 with no sign of weakening. This unprecedented freshening was caused by preceding extreme Arctic sea ice melt events[24].

Reduced winter cooling, enhanced summer warming, endured upper layer freshening, and shoaled winter convection, through either positive (direct) *thermal expansion* or negative (inverse) *haline contraction*, all increase the depth-integrated specific volume and thus steric height of the CLS. (*Haline contraction refers to the decrease in volume of a mass unit of seawater in response to an increase in its salinity under constant temperature and pressure. Inversely, freshening causes expansion or negative contraction.*) We hypothesize that these concurrent oceanographic processes have collectively elevated the CLS sea level to a record high. We test this hypothesis and its applicability by addressing the following eight questions:

[1] What fraction of the altimetry-based sea level variations is explained by the steric height changes derived from hydrographic observations?

[2] To what extent does winter convection, dominating hydrographic variability in the intermediate, 200–2000 dbar, layer[24], influence steric height and, consequently, sea level?

[3] Does the halosteric contribution to the interannual steric height changes always act to counterbalance the thermosteric contribution, as stated previously[37–40]?

[4] Alternatively, was there any extended recent time period, comprising several years, over which the halosteric contribution kept continuously reinforcing the thermosteric contribution, instead of counterbalancing (i.e., offsetting) it?

[5] If the latter is true, then how necessary was it for the halosteric contribution to reinforce the thermosteric one in order to achieve a record high sea level in one of the recent years?

[6] What is the contribution of the deep (2000–3700 m) layer[24], beyond the core Argo float profiling range, to the steric height changes?

[7] How accurately can the thermosteric height changes be reproduced by utilizing the atmospheric data?

[8] How does the halosteric height respond to an increased Arctic sea ice melt?

These questions and their sought-after answers carry important implications. In particular, an affirmative answer to the fourth question would underscore the critical importance of monitoring regional freshwater content (i.e., salinity) year-round, and the multiyear cycles of deep convection for both operational and long-term forecasting needs. The deep steric contribution, especially in regions where deep-water forms, has rarely been quantified from a time series perspective due to limited data availability. To address these questions, we examine how various local and remote hydrographic, sea ice, and atmospheric processes contribute to the prominent *ship and Argo float-based* steric height anomalies (SHA) and *satellite altimetry-based* sea level anomalies (SLA).

This study primarily focuses on interannual changes in CLS SLA and SHA during 1992–2025, a period marked by the most considerable recorded changes in heat content, freshwater accumulation, and

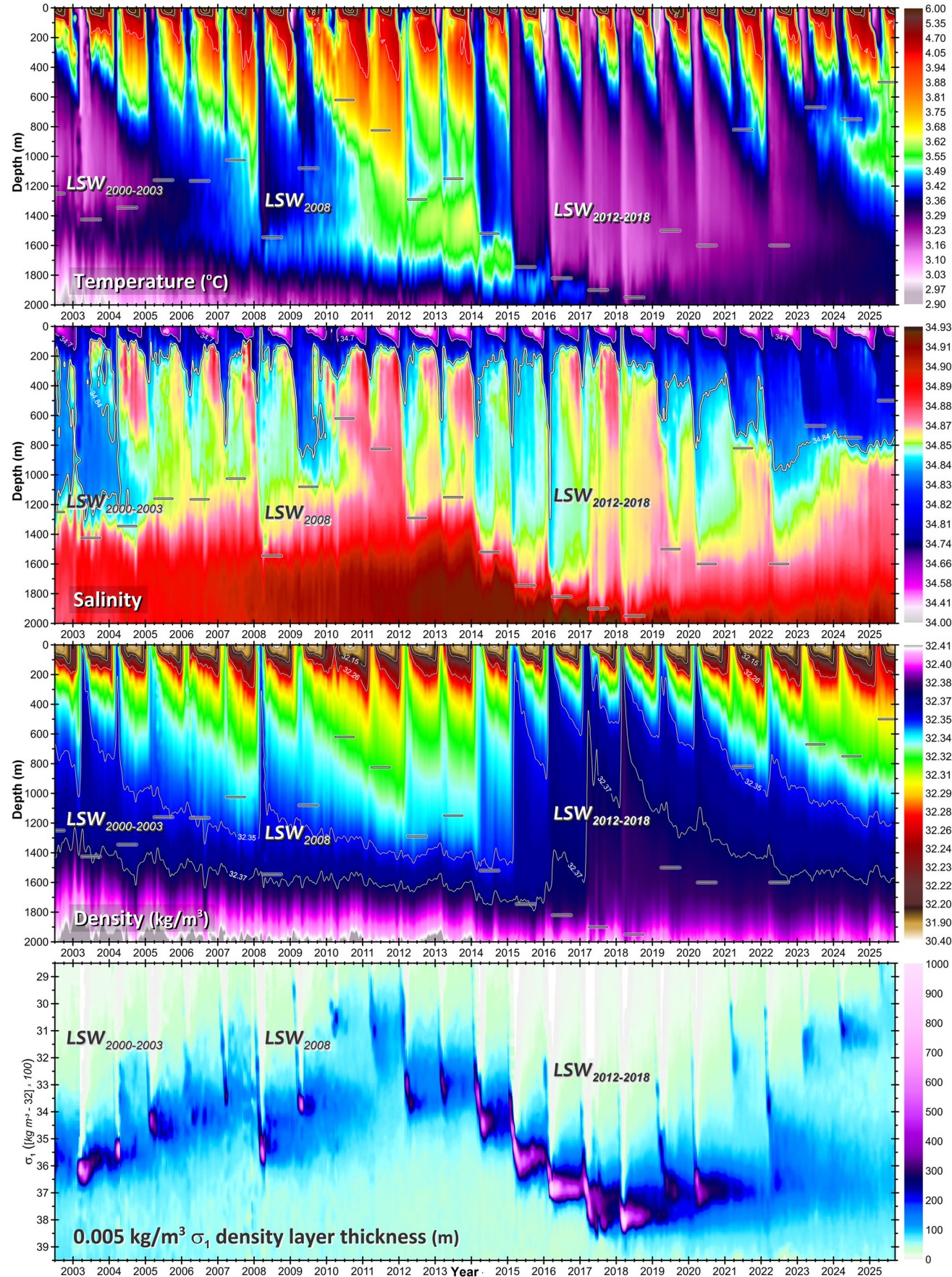

**Fig. 2 | Variability of the central Labrador Sea temperature, salinity, density, and 0.005 kg/m³ density layer thickness from 2002 to 2025.** The vertical profiles used in the figure are based on weekly-to-biweekly binning of quality-controlled and calibrated Argo float and ship-based observations. Short horizontal lines indicate the convection depths. LSW subscripts denote year classes.

winter mixing depth, and coinciding with the satellite altimetry era, and in CLS SHA starting from 1948. By comparing the long-term trends, multiyear cycles, and extremes of SLA, SHA (as well as its thermosteric and halosteric components), and associated hydrographic properties, we investigate the underlying origins and drivers of these changes by sequentially addressing *the first-to-fifth questions*. The SLA and SHA changes concurrently resolved since 1992 are then placed in the context of SHA variability over the period of 1948–2025. The key goal of this retrospective analysis is to determine how recent conditions differ from those of the past, particularly with respect to the third-to-fifth questions. Furthermore, we assess the contribution of certain vertical layers to the total steric height changes, with particular emphasis on the role of the deep ocean, as asked in the sixth question. Finally, we discuss how air-sea heat exchange and Arctic sea ice meltwater contribute to regional heat and freshwater content changes, and how this knowledge can help us to accurately model thermosteric and halosteric variations, and hence sea level trends, thus addressing the last two questions.

## Results

Our SHA analysis is exclusively based on the original (i.e., unaveraged and ungridded spatially and temporally) Argo float and ship-based hydrographic measurements. Analogously, the along-track altimetry data is used to construct the CLS SLA time series. As part of cross-validation, the CLS SLA time series derived from along-track and gridded altimetry (Supplementary Figs. 6 and 7) show similar patterns across seasonal, interannual, and multidecadal timescales throughout the record.

The raised research questions are addressed by studying the interannual changes, trends, and extremes of SLA, SHA, and thermosteric and halosteric heights. Prior to analysis, the respective climatological seasonal cycles are subtracted from individual measurements, which are then filtered, smoothe,d and bin-averaged, as described in the "*Methods*" section.

### Climatological seasonal cycle of sea level and its components

Since we analyze anomalies, computed as deviations from the respective climatological seasonal cycles, we show these cycles in Figs. 3 and 4, and Supplementary Figs. 6–10 to facilitate comparisons and summations with the anomaly-derived values (Supplementary Figs. 9 and 10 have seasonal cycles added to anomalies).

The distinct seasonal cycles of the detrended SLA and SHA have considerably different magnitudes (Fig. 3). This difference is highly insightful for the regional freshwater and ocean dynamics, as it explicitly points at a summer mass gain, supported by satellite gravimetric data (Fig. 3, "*Methods*" section), further discussed in the "*Remaining challenges and future steps*" subsection.

The thermosteric component dominates the seasonality of SHA (Fig. 4 and Supplementary Fig. 8). The specific volume, temperature, and salinity, and steric, thermosteric, and halosteric height seasonal cycles examined across the 10–900 dbar pressure range (Fig. 4) reveal the pressure-dependence of the respective seasonal amplitudes and phases. While the thermosteric amplitude monotonically decreases with pressure, the overall smaller halosteric amplitude reaches its maximum within the 100–200 dbar range. While the temperature and thermosteric height phases monotonically increase with depth, the salinity and halosteric phases reverse between 10 dbar and 150 dbar.

### Extreme sea level observed in the central Labrador Sea in 2025

In the CLS, SLA has varied significantly over the satellite altimetry era (1992–2025), reaching a record low in 1994, a notable local minimum in 2017, and a record high in 2025 (Fig. 5). The full-depth SHA, which shows similar interannual variability, is constructed as explained below.

Since the core Argo profiling depth is limited to 2000 dbar, there are no sufficient measurements to resolve sub-annual hydrographic changes in the deeper layer. Fortunately, the CLS water column below 1500 dbar is dominated by interannual variability (and the deeper the stronger, Fig. 2). Therefore, to sufficiently cover the entire water column to resolve the prevailing timescales, we analyze the 1900–3300 dbar layer separately from the overlaying, 10–1900 dbar layer. This approach facilitates a focused assessment of the deep-layer SHA interannual variability and long-term trends, providing a baseline study for measuring and understanding the contribution of the deep layer to the steric height changes. To represent the full-depth, 10–3300 dbar, water column in our research, we combine the upper, 10–1900 dbar, and deep, 1900–3300 dbar, layer-based estimates.

While having measurably different seasonal cycles (Fig. 3), SLA and full-depth SHA show strikingly similar interannual changes, trends, and extremes. This similarity is evidenced by a high, 0.985, correlation of the yearly averaged SLA and SHA, hereafter called YASLA and YASHA, respectively and a low, 0.67 cm, standard deviation of their difference. Detrending of YASLA and YASHA does not affect the strength of their similarity, giving, respectively, 0.960 and 0.67 cm for the noted metrics. Furthermore, YASHA accounts for approximately 97% of the interannual variability of YASLA before detrending, and 92% after detrending, with individual values within 1.4 cm of each other in 97% of all cases, both before and after detrending. In 2025, both YASLA and YASHA reached record highs, signified by the respective squares in Fig. 5 standing 18.4 cm and 15.0 cm higher than 31 years earlier, at the dawn of the satellite altimetry era. The 34-year, 1992–2025, trends of YASLA and YASHA, $0.308 \pm 0.007$ and $0.300 \pm 0.007$ cm/year, respectively, which small difference of ~2.6% is attributed to the water-column mass change. Coincidentally, a 30-year period is commonly regarded as a baseline for climatological normal[42], signifying this study as the first-of-its-kind climatological assessment of sea level changes in the CLS.

The strong alignment of SLA and SHA underscores the fact that sea level variability in the CLS is essentially (i.e., by 97%) steric, and is driven primarily by full-depth variations of temperature and salinity. *This answers the first key question.*

### Winter convection regime changes drive sea level variability

Figure 2 provides insight into the role of the multiyear convective cycles in defining and shaping the SHA and, consequently, SLA trends and extremes (Fig. 5). Thermosteric and halosteric components of SHA, shown in Fig. 6, quantify the respective roles of the full-depth temperature and salinity changes driven by both heat and freshwater fluxes and convection.

The answer to the first key question in the previous subsection is supported by the correlation between the altimetric sea level (SLA and YASLA) and the total steric height for the entire water column (SHA, YASHA), and associated statistics. Since all analyzed years, except 1992–1995, the direct effect of winter convection is restricted to the top 2000 db (Fig. 2), in this subsection, we provide analogous statistics for the 10–1900 dbar layer steric height instead of the full-depth one. The correlation and standard deviation of the difference of YASLA and 10–1900 dbar YASHA are 0.961 and 1.08 cm before detrending, and 0.932 and 0.95 cm after detrending, respectively. YASHA explains 92% of the interannual variability of YASLA before detrending and 87% after detrending.

The aforementioned convective intensification-to-relaxation phase change, evident in Fig. 2, had a direct impact on YASHA and YASLA (Figs. 5 and 6 for full-depth YASHA, and Supplementary Figs. 11 and 12 for 10–1900 dbar YASHA), manifesting in their respective trend changeovers around 2018. Stronger convective mixing events produce denser thicker pycnostads (Fig. 2, *bottom panel*), and, in turn, lower SHA and SLA. In contrast, weaker convections produce lighter, thinner pycnostads, stronger vertical stratification, and, in

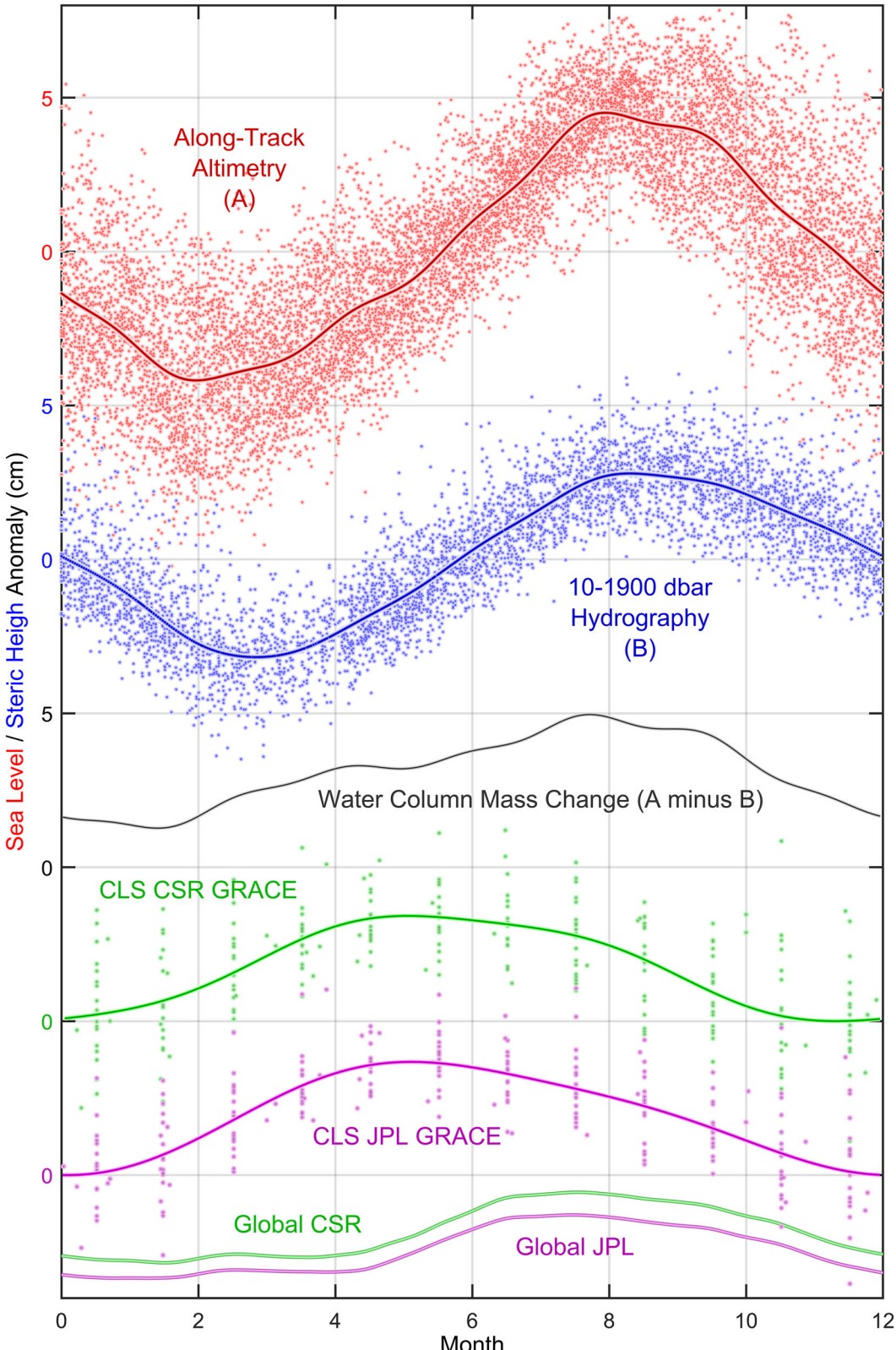

**Fig. 3 | Seasonal cycles of the total sea level, and its steric height and water-column mass components in the central Labrador Sea.** *Top-down* (cm): seasonal variability (*dots*) and regular or climatological seasonal cycles (*lines*) of the central Labrador Sea total sea level derived from satellite along-track altimetry data with outliers and interannual variations mostly removed (*red, labeled "A"*), and its 10–1900 dbar steric height component based on individual Argo float and ship survey observations with outliers and interannual variations mostly removed (*blue, labeled "B"*, the regular seasonality weakens with pressure, e.g., Figs. 2 and 4, justifying the 1900 dbar cutoff); water-column mass component computed as the difference of the total sea level (A) and steric height component (B) regular seasonal cycles (*grey/black, labeled "A minus B"*); and water-column mass components based on CSR (*green*) and JPL (*purple*) GRACE datasets (*dots*); and the global ocean (deeper than 2000 m) CSR (*green*) and JPL (*purple*) mass seasonalities.

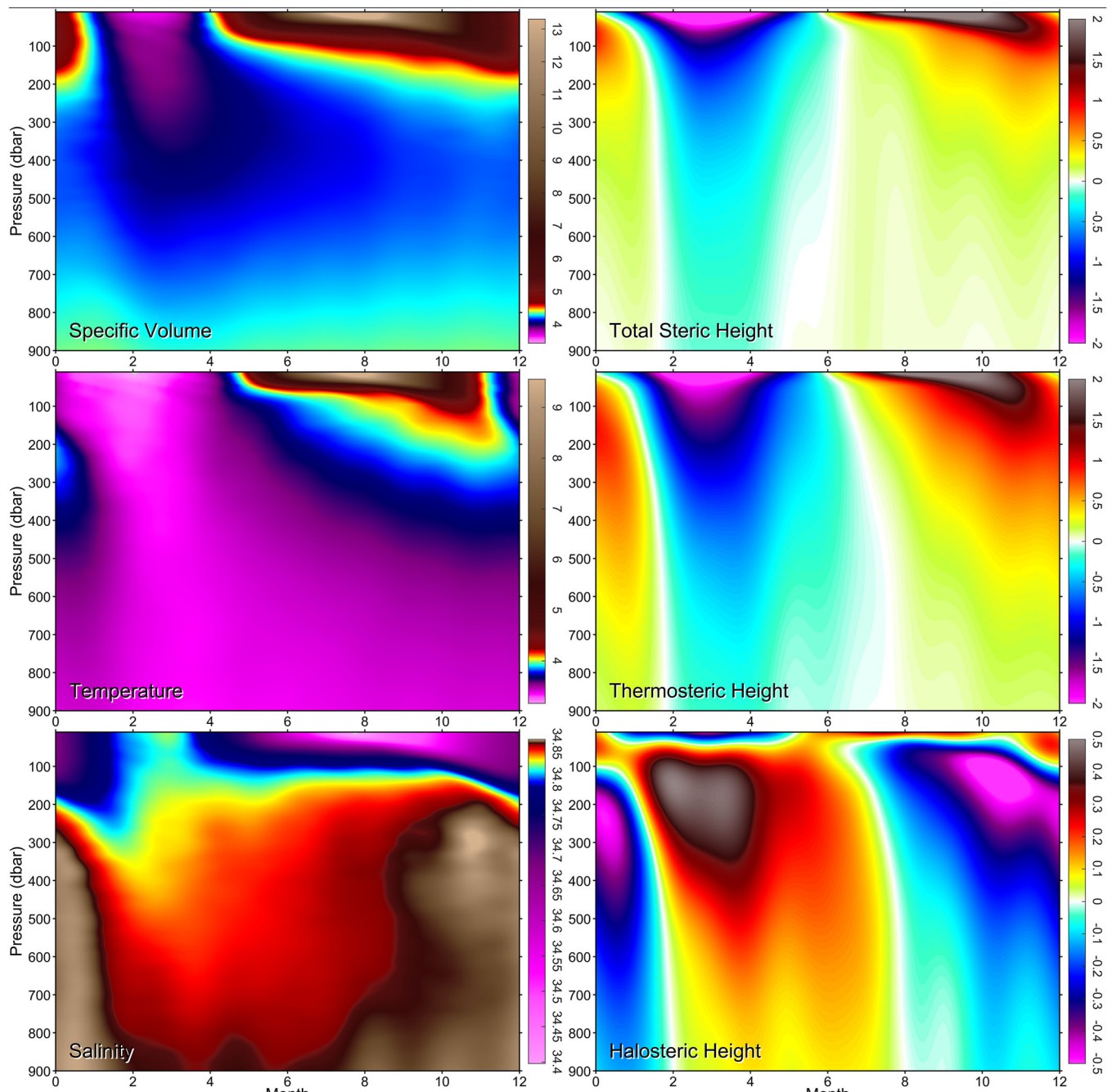

**Fig. 4 | Climatological seasonality of hydrographic and derived variables across the top 900 dbar layer of the central Labrador Sea.** The regular seasonal cycles of specific volume, temperature, and salinity (*left*), and steric, thermosteric, and halosteric heights (*right*) based on the quality-controlled and calibrated Argo float and ship-based measurements collected in the central Labrador Sea during 2002–2025, and referenced to 1900 dbar.

turn, higher SHA and SLA. This explains why the main driver of convection and deep-ocean cooling—net winter surface heat loss—is also in control of YASHA and YASLA through ocean cooling dominating mixed layer density, convection depth, and vertical stratification.

Summarizing the material presented in this subsection, we conclude that recurrent winter convection, with its 2010–2011 and 2021, 2023–2025 lows, and 1994 (*discussed later*) and 2018 (Fig. 2) highs, through the thermosteric height, dominates YASHA (Fig. 6). *This answers the second key question.*

On the other hand, driven by the massive upper-layer freshening, the effect of the year-to-year halosteric height changes (our next focus) has recently switched from moderating (i.e., reducing) to reinforcing (i.e., amplifying) the SHA and, consequently, SLA upward trends,

earlier influencing the timing of the SHA/SLA fall-to-rise reversal, shifting it from 2018 to a slightly earlier date.

**Reversal of the upper layer's temperature-salinity relationship accelerated the sea level rise**

The yearly averaged full-depth halosteric height resides within about half the thermosteric height range (Fig. 6, squares). Besides, the interannual halosteric and thermosteric height changes are often opposite in sign, with the former tending to counterbalance the latter. Can it be, therefore, concluded that the sea level changes driven by thermal expansion are generally half-compensated by concurrent haline contraction? Apparently, it can. Indeed, according to Fig. 6, nearly every year from 1990 to 2015, the full-depth halosteric

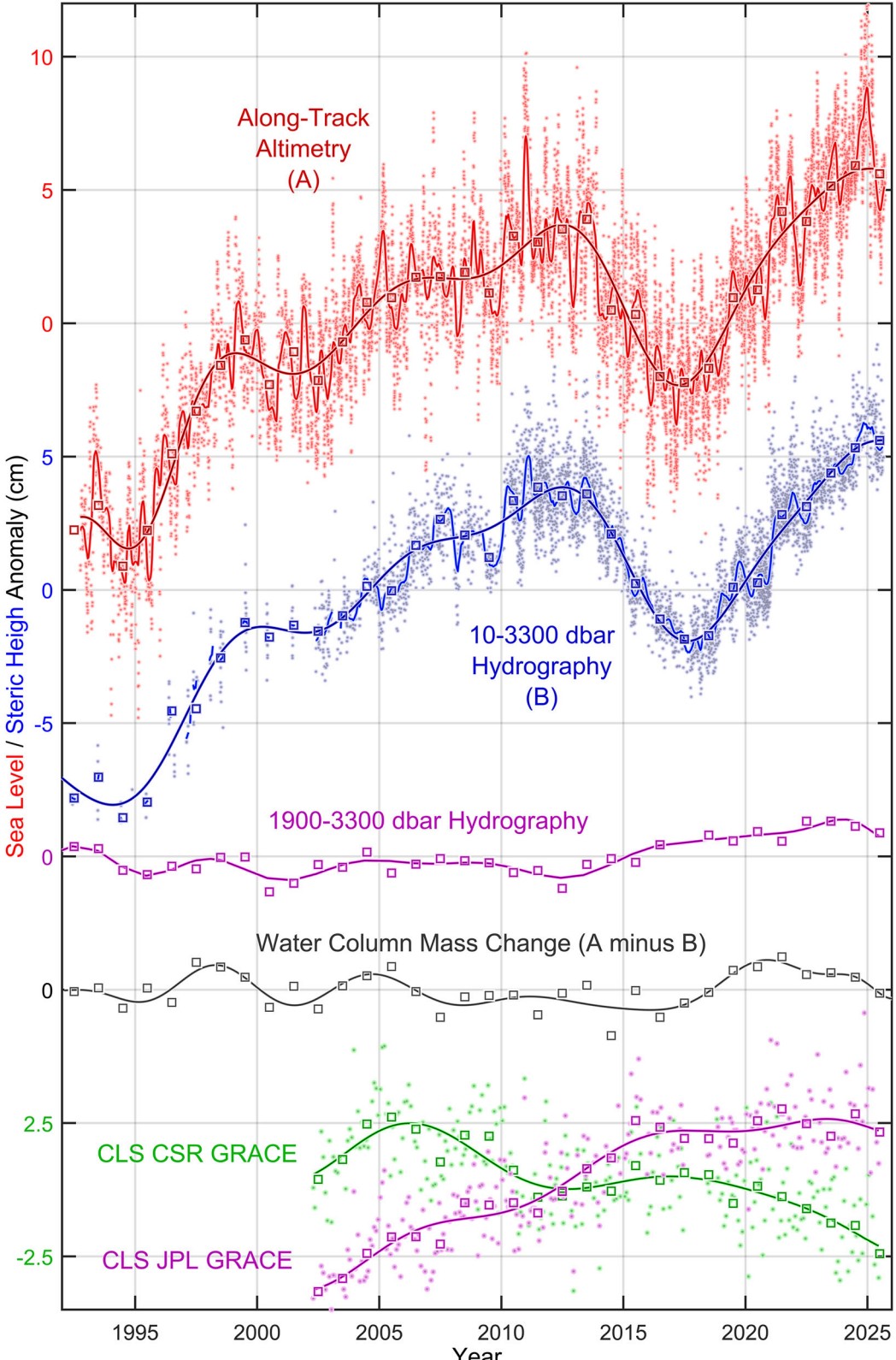

**Fig. 5 | Variability of the total sea level, and its steric height and water-column mass components in the central Labrador Sea from 1992 to 2025.** Top-down (cm): the along-track satellite altimetry-based total sea level (red, labeled "A"); its hydrography-based full-depth (blue, labeled "B") and deep-layer (purple) steric height components; and three independently-derived water-column mass components. The first mass component (grey/black, labeled as "A minus B") is computed as the difference of the yearly-averaged total sea level (A) and full-depth steric height (B) values. The other two mass component are based on the CSR (green) and JPL (purple) GRACE data. Outliers and regular (climatological) seasonal cycles have been removed from all original time series (dots) through an iterative process described in the "Methods" section. Squares, underlined with optimal polynomial fits, indicate the yearly averaged residuals. The summed irregular seasonal and low-frequency signals of (A) and (B) are shown with lighter lines.

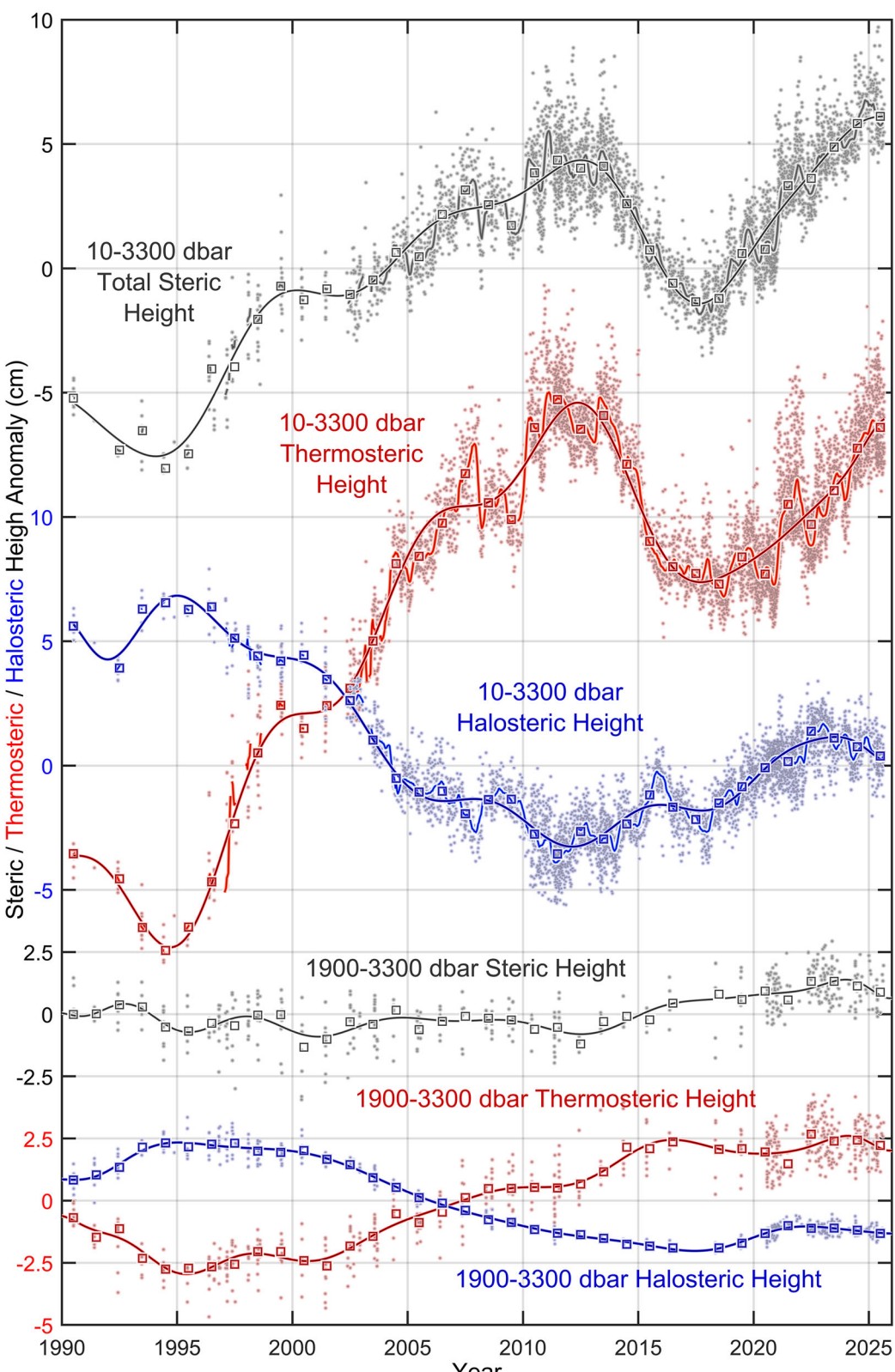

**Fig. 6 | Variability of the central Labrador Sea total steric, thermosteric, and halosteric heights from 1990 to 2025.** *Top-down* (cm): the full-depth (10–3300 dbar) steric (*gray/black*), thermosteric (*red*) and halosteric (*blue*), and deep-layer (1900–3300 dbar) steric (*gray/black*), thermosteric (*red*) and halosteric (*blue*) heights based on the original Core and Deep Argo float, and shipboard hydrographic observations. Outliers and respective regular (climatological) seasonal cycles have been removed from the original time series (*dots*) through an iterative process described in the "*Methods*" section. *Squares*, underlined with optimal polynomial fits, indicate the yearly averaged residuals. The summed irregular seasonal and low-frequency full-depth height signals are shown with *lighter lines*.

component consistently counteracts the thermosteric component, offsetting year-to-year thermosteric height changes by about a half. Such persistence of the halosteric–thermosteric counterbalancing showing through 2015 might imply that the partial compensation of thermal expansion by haline contraction would recur, if the situation had not radically changed after that year.

In 2016, the halosteric contributions to the year-to-year steric height changes switched from counterbalancing to reinforcing the respective thermosteric contributions. *This answers the third key question*. This transition arose as the halosteric height maintained a positive trend throughout the 2011–2025 period, while the thermosteric height rebounded from its decline in response to the 2012–2025 convective cycle's phase change (Fig. 2). The physical processes responsible for the recent transition from halosteric–thermosteric counterbalancing (before 2016) to their allying and covarying (2016–2025) become evident from Fig. 2, Supplementary Figs. 1–4, and the published analysis of the relevant signals and their causes[24] recapped here. The CLS water column steadily cooled as convection deepened between 2011 and 2018. Then, these trends reversed as convection entered a relaxation phase. In turn, salinity, unlike temperature (Fig. 2), and thus the halosteric component, unlike the thermosteric component (Fig. 6), exhibited distinct steady multiyear trends throughout the entire 2011–2025 period, with upper-layer freshening being interrupted only briefly in 2016 and 2017 without disturbing the overall trends. *This answers the fourth key question*.

As shown in Fig. 2, and Supplementary Figs. 1–4, and is further illustrated and explained in the last, "*Linking halosteric height changes to the extreme Arctic sea ice losses*", subsection of the "*Results*" section, in 2025, the upper (15–100 m) and upper intermediate (300–500 m) layer salinities reached record lows, directly caused by advection of anomalous quantities of freshwater produced by extreme Arctic sea ice melt a few years earlier[24], while the full-depth halosteric height reached its maximum somewhat earlier, during 2022–2023 (this shift is explained in the noted subsection). Driven by the unbalanced freshwater inflow into the Labrador Sea, the halosteric height increased by approximately 5 cm between 2011 and 2023. In contrast, the thermosteric height was lower in 2023 than in 2011, as post-2018 warming could not fully compensate (and thus offset) the 2011–2018 cooling. Overall, the contribution of salinity has amplified that of temperature after winter convection entered the relaxation phase, making the sea level to be higher in 2023 than in 2011 (Figs. 5 and 6). The low-salinity state of the upper 800 m layer acquired in 2023 persisted through most of the two following years, 2024 and 2025, keeping the halosteric height relatively high, although it slightly declined toward the record's end (Fig. 6). Concurrently, between 2023 and 2025, the further shoaling of winter convection to record shallow and sea warming sustained the upward trends of SHA and SLA, which reached record highs in 2025. The sea level rise between 2023 and 2025 was mostly due to the thermosteric component.

Overall, in contrast to moderating (i.e., reducing or partially offsetting) the density responses to both positive and negative temperature changes prior to 2016, the salinity changes of the subsequent years amplified these responses, aiding first the convection-driven downward and then upward thermosteric trends. By reinforcing instead of compensating the thermosteric height changes in the latter years, the halosteric component boosted the temperature-driven sea level rise, creating a marked extreme. Indeed, while through 2023–2025 the thermosteric height stayed 3.6–1.0 cm below its record high that was registered in 2011 (Fig. 6, red squares), the accentuated halosteric–thermosteric relationship changeover from counter-directional to codirectional played a critical role in the CLS sea level rise to a record high in 2025. *This answers the fifth key question*.

Moreover, record-breaking rates of the YASHA and YASLA changes over a sliding eight-year interval also fall on 2017–2024. These rates surpass even those for the 1994–2001 period brought up by a rapid

recovery from record cold, fresh and dense conditions, alongside record low YASHA and YASLA levels (Figs. 5 and 6). A fundamental difference between these two periods lies in the behavior of the halosteric height: in 1994–2001, it counterbalanced about half of the thermosteric height change, whereas in 2017–2023, it reinforced the other, accelerating the total SHA and hence SLA increase.

## The deep layer reinforced the recent sea level trend

The role of the deep layer in the recent SLA trend and extreme becomes evident when comparing the full-depth YASHA (Fig. 5) with that of the upper 1900 dbar layer (Supplementary Figs. 11 and 12). While the 2012 and 2025 upper-layer YASHA were not significantly different, the full-depth YASHA was at a record high in 2025. After two decades, 1992–2011, of relative stability with rather small fluctuations (± 0.6 cm) deep-layer YASHA underwent a positive trend of ~0.22 cm/year, implying a cumulative deep SHA increase of ~2.4 cm throughout 2012–2023 (Figs. 5 and 6). This change exceeds the ~0.9 cm increase in full-depth YASHA over the same period, emphasizing the critical role of the deep layer in raising full-depth SHA, and, consequently, SLA, to their unprecedented highs reached in 2025.

The origin of the positive 2012–2023 deep-layer YASHA trend is revealed through a decomposition of the deep-layer YASHA series into the respective thermosteric and halosteric components and their close co-examination (Figs. 6 and 7). Between 1990 and 2011, these components recurrently rebalanced each other, effectively suppressing development of any significant decadal trend in deep-layer YASHA (Figs. 5 and 6). This delicate equilibrium was irreversibly disrupted in 2012, causing a shift in the thermohaline balance that persisted for eleven consecutive years. Throughout this period, the stronger variability of the deep thermosteric height compared to that of the deep halosteric height dominated the deep-layer YASHA; consequently, a positive trend. *This answers the sixth key question*.

## Did a halosteric reinforcement of a thermosteric trend happen in the past?

Recognizing that the halosteric reinforcement of the thermosteric height changes occurring over the period of 2016–2023 played a critical role in the recent sea level rise to a record high (Fig. 6), we extend our analysis back to 1948 in order to find out if similar events ever occurred before 1996.

The ship and Argo float-based hydrographic measurements collected in the CLS from 1948 to 2023 have provided insights into multidecadal cycles of winter convection, water-column cooling and warming, freshening, and salinization[24]. To investigate the impacts of these changes on the regional steric height components, we, upon quality-checking and editing all in situ observations, construct for each year of the 1948–2025 period, meeting the data sufficiency requirement, composite vertical profiles. The steps of this data synthesis are outlined in the "*Data Sources*" and "*Methods*" sections, and our earlier publications[24]. The resulting full-depth annual profiles, compiled in Supplementary Fig. 1, fully and accurately capture all major hydrographic developments in the Labrador Sea, including the multiyear convective cycles[24], since the late 1940s. To optimize quality and comparability of the in situ observations entering the underlying *annual state-of-the-ocean profile making process* (and hence Supplementary Figs. 1 and Fig. 7), here, the Argo and, when available, Deep Argo profiles only fill in for missing or insufficient ship-based observations (all Deep Argo float profiles have been checked and calibrated to achieve 0.002 °C temperature and 0.002 salinity accuracies as explained in the "*Supplementary Information*" section of the mentioned study[24], like in 2017 and 2021, while being left out otherwise. The annual profiles are then used to derive the YASHA time series and its components across pressure levels, spaced at 5 dbar intervals from 200 to 3000 dbar and referenced to 3300 dbar, as shown in Fig. 7 (the 3000–3300 dbar layer is not displayed as changes are weak there).

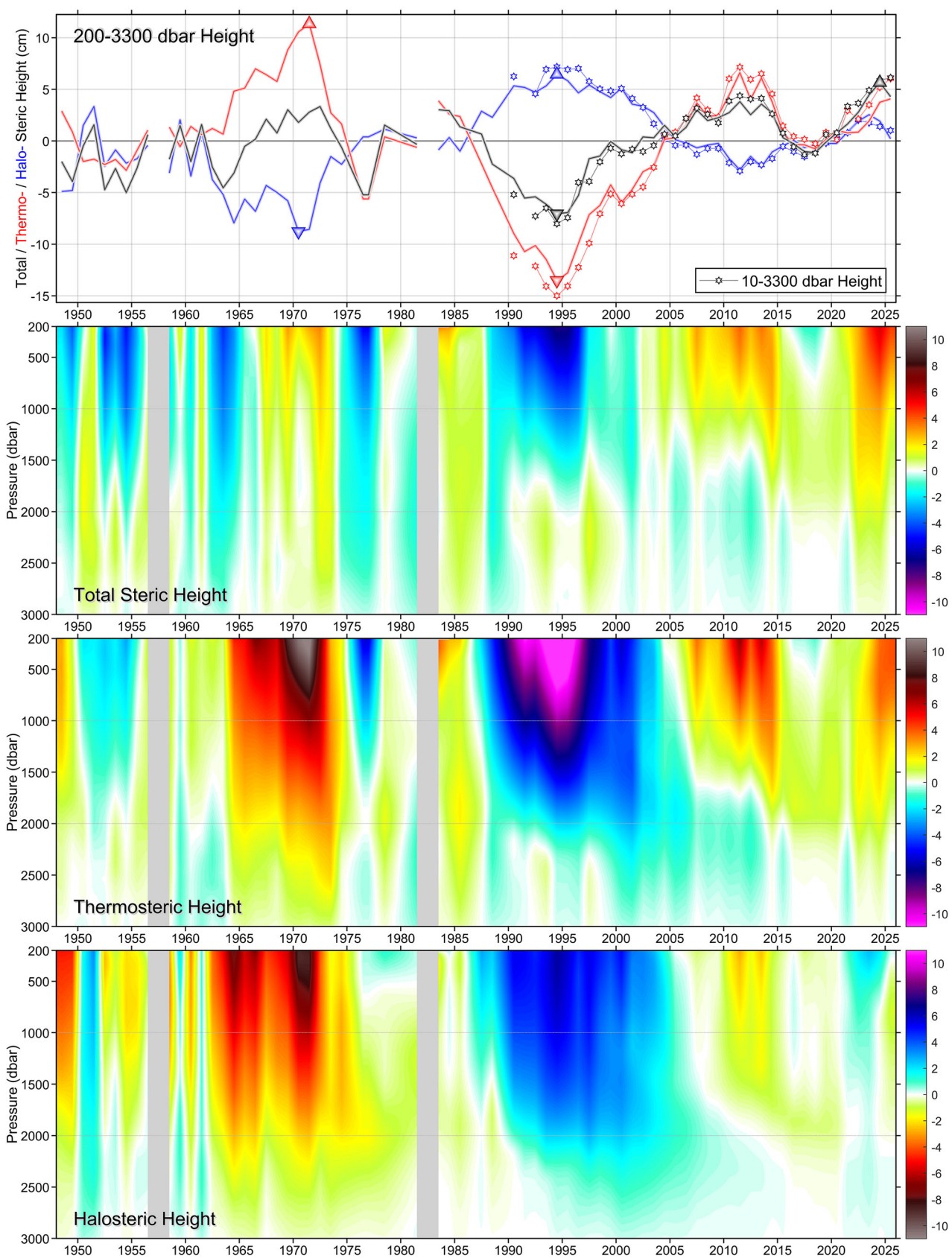

**Fig. 7 | Evolution of vertical profiles of the central Labrador Sea steric, thermosteric, and halosteric height from 1948 to 2024.** *Top-down*: the 1990–2025 10–3300 dbar all-inclusive hydrography-based and 1948–2024 200–3300 dbar ship hydrography-based total steric, thermosteric and halosteric (*gray/black, red and blue, respectively*) height anomalies; and evolutions of the 200–3300 dbar total steric, thermosteric and halosteric height yearly averaged profiles (*trimmed to 3000 dbar as the deeper changes are weak*).

As seen in Figs. 2 and 4, the seasonal cycle dominates temporal variability in the top 200 dbar layer (e.g., the seasonal cycle accounts for >92% of the total temperature variance at 10 dbar). Furthermore, even with the regular seasonal cycle subtracted from infrequent pre-Argo (1948-2002) ship-based measurements, undersampled irregular seasonal variations may still be present there, contributing spurious signals to yearly averaged values. The upper 200 bar (meter) layer data, prone to residual seasonal aliasing, have been excluded from the pre-2003 compilations shown in Fig. 7. This exclusion has an insignificant effect on the full-depth, 10-3300 dbar, steric, thermosteric, and halosteric heights. The corresponding 10-3300 dbar and 200-3300 dbar height differences, particularly for the steric height, are much smaller than the decadal changes (Fig. 7, *top*), justifying our choice of the 200-3300 dbar heights for analyzing longer-term variations in CLS steric sea level. Furthermore, while the exclusion of the upper 200 dbar layer slightly reduces the range of height changes, the patterns remain unaffected.

The 200-3300 dbar steric, thermosteric, and halosteric height extremes for the 1948-2025 period are highlighted with triangles in Fig. 7 (*top*). The timing of these events suggests the following:

[1] A thermosteric high and a halosteric low occurred between 1970 and 1971.

[2] Conversely, a thermosteric low and a halosteric high were recorded in 1994.

[3] The total steric height reached its absolute minimum concurrent with the lowest thermosteric and highest halosteric values, whereas its absolute maximum, achieved in 2024, is comparatively less pronounced an extreme in either component.

[4] While all major hydrographic events (e.g., anomalies, trends, mixing events), occurring in the CLS between 1948 and 2015 had their sustained thermosteric signals counterbalanced and significantly compensated by the respective halosteric signals, the halosteric and thermosteric changes during the 2015-2023 period were of the same sign and had comparable magnitudes, reinforcing each other in SHA and SLA. Notably, the positive coupling of the post-2015 thermosteric and halosteric height changes, coinciding with the most substantial freshening of the upper 700 dbar layer (Fig. 2 and Supplementary Figs. 1-4), challenges the conventional vision of CLS steric height changes based on the assumption of counteraction of halosteric and thermosteric changes and trends through density compensation. It also emphasizes the role of changes in horizontal advection of heat, salt, and, especially, freshwater in accelerating or moderating the sea level rise.

Between 1953 and 1956, we see another instance of the two steric components changing in the same direction. However, taking into consideration the short duration of that event, the sparseness of water sample observations, and the varying accuracy and resolution of the hydrographic measurements of that time, we refrain from making any conclusive statement about the possibility of a halosteric effect reversal during that time *(the mentioned sparseness, low quality and low quantity of salinity measurements is reflected in* Supplementary Fig. 1 *in the patchy vertical strips between 1948 and 1975. In contrast, the period of 1990-2019 features uniformly high quality and sufficiency of hydrographic observations, cataloged in Supplementary Table).*

Figure 7 also provides detailed insight into the intermediate and deep layers' contributions to YASHA variability. YASHA time series constructed for pressure levels spaced at 5 dbar are brought together in the *second panel from the top*. This compilation clearly shows where in the water column, when, and how fast each trend developed, weakened, and reversed, and which layer shaped it the most. In most cases, the interannual changes and trends tend to reverse anywhere between the pressure levels of 1250 dbar and 2000 dbar. These reversals, regardless of their exact vertical positions, explain why the trends observed at 200 dbar and 2250 dbar are so profoundly different. Over

the 1990-2021 period, the 200 dbar and 2250 dbar YASHA series displayed opposing trends, each punctuated by short-term reversals in the opposite direction. This inverse symmetry of YASHA changes between the upper and deep layers suggests a counterbalancing effect, where upper-layer changes are offset by those in the deep layer, or vice versa. What is particularly important to our study is that in 2022-2024, unlike the previous years, YASHA were consistent across all pressure levels, indicating a positive contribution from the deep layers to the 200-3300 dbar and full-depth steric heights.

Further insights into the impact of the deep layer on upper ocean SHA changes are obtained by comparing thermosteric and (reversely color-coded for easier comparison) halosteric component variations with depth (Fig. 7). Below 1500 dbar, the two components largely counterbalance each other, although the resulting compensation is not full at all times. For instance, the halosteric component dominated deep-layer residuals from 1991 to 1999, while the thermosteric component did so from 2005 to 2016. In 2022-2024, both deep halosteric and thermosteric height anomalies were positive, amplifying the upper-layer signal significantly.

Unlike its deeper counterpart, the water column's segment above -1500 dbar, throughout its extent, is typically strongly affected by the thermosteric component as the upward halosteric gain remains relatively weak. This promotes faster accumulation of thermosteric anomalous signals toward the surface and their prevalence in full-depth YASHA. Strikingly, here again, 2022-2024 were exceptional. In these years, the upward halosteric gain matched the thermosteric one.

The provided explanation of the deep and intermediate layer contributions to the exceptional CLS sea level rise during 2022-2025 underscores the uniqueness of the event and raises its importance for the subpolar and larger North Atlantic domains, as local deep and full-depth regime shifts are likely to affect broader-scale ocean dynamics and exchanges.

## Can the thermosteric height changes be reproduced using atmospheric data alone?

The sea level of the central Labrador Sea (CLS) reached a 78-year record high in 2025. The rapid sea level rise that led to this event was caused by the joint action of mild winters, warm summers, sustained shutdown of deep convection, exceptional upper ocean freshening, deep-ocean halosteric–thermosteric balance shift, and full water-column mass gain. While both deep, 1900-3300 dbar, layer steric height and full-column mass changes contribute to the long-term sea level trend, the interannual-to-decadal variability is predominantly shaped by the upper, 10-1900 dbar, layer thermosteric and halosteric components. Our understanding of the interannual-to-decadal sea level changes is further advanced following the approach presented here for the reconstruction and prediction of the two steric height components. We first demonstrate how realistically detailed and accurate our pseudo thermosteric heights are simulated using the atmospheric forcing data (Fig. 8), and then discuss the predictability of the halosteric height.

The thermosteric component dominates both seasonal (Fig. 4 and Supplementary Fig. 8) and interannual (Figs. 6 and 7) variabilities of SHA, and, consequently, those of SLA (Figs. 3 and 5). The seasonal patterns of temperature and thermosteric height change their shapes with pressure (depth) (Fig. 4), revealing two distinct vital signals with unique vertical penetration routines, pressure-dependent time lags, and appearances—a deeper winter cooling signal and a shallower summer warming signal. The winter cooling is regulated by *Winter Surface Heat Loss* (WSHL), which was thoroughly analyzed in ref. 24 and is calculated by integrating all components of surface heat budget over an individually-defined cooling period[24]. In contrast, *Summer Surface Heat Gain* (SSHG) is calculated for the period when the accumulated net heat flux is directed into the sea. Another characteristic of seasonal warming, used in our work, is

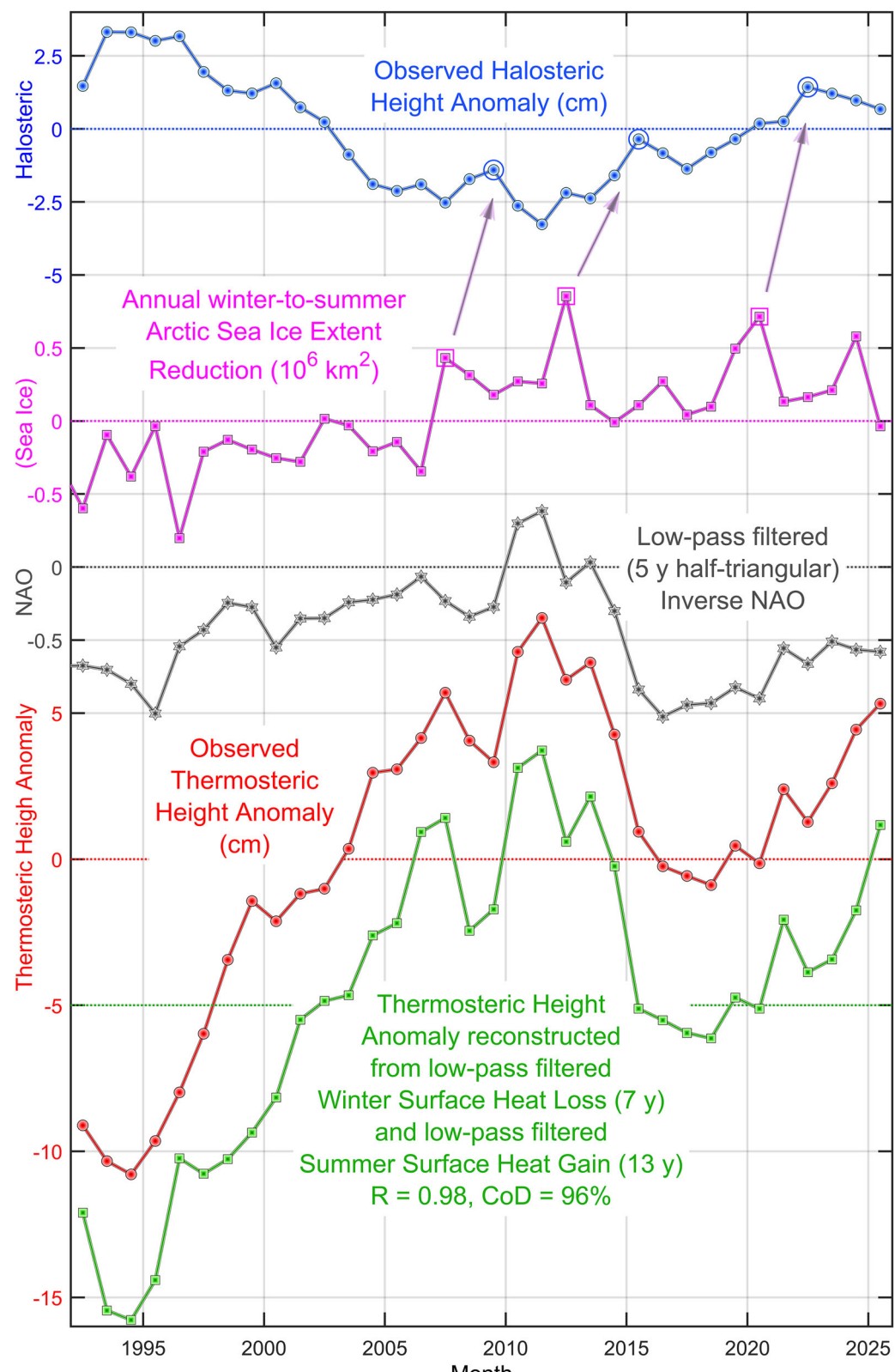

**Fig. 8 | Reconstruction and prediction of the central Labrador Sea thermosteric and halosteric heights using the atmospheric forcing indices and annual Arctic sea ice losses.** *Top-down*: the observed 10–1900 dbar halosteric height anomaly (cm, *blue*), winter-to-summer Arctic sea ice extent reduction ($10^6$ km², *purple*), winter (DJFM) North Atlantic Oscillation (NAO) index (inverted, *gray*), and observed (cm, *red*) and reconstructed (from winter surface heat loss and summer surface heat gain, cm, *green*) 10–1900 dbar thermosteric height anomalies. Arrows connect extreme winter-to-summer Arctic sea ice losses to local halosteric maxima (highs). The method of thermosteric height reconstruction is explained in the "Methods" section and Supplementary Fig. 13 caption. The local halosteric height maxima of 2009, 2016 and 2022 can be linked to the extreme Arctic sea ice losses of 2007, 2012 and 2019–2020, respectively as shown in Fig. 9.

*Summer Heat Peak* (SHP). SHP is averaged over a fixed-length time interval (e.g., 15, 21, 31 days) centered on a daily surface heat gain peak. The surface heat exchange characteristics are further detailed in the "*Seasonal air-sea heat exchange metrics*" subsection and Supplementary Fig. 13 caption.

To empirically model the observed yearly averaged thermosteric height changes we make three key assumptions:

[1] air-sea heat exchange is the leading factor controlling the thermosteric height,

[2] the interannual changes of WSHL and SSHG have different in magnitude and differently lagged effects on the thermosteric height, and need to be assessed separately, and

[3] the residual impacts of previous cooling and warming events (preconditioning[24]) can be approximated by some asymmetric low-pass filtering (e.g., autoregressive; or, here, with left-side half-triangularly weighted window).

We reconstruct the yearly averaged thermosteric height series by optimally low-pass filtering, scaling, and merging WSHL and SSHG or, alternatively, SHP (SSHG|SHP). All sought parameters are found through iterative approximations aiming to minimize either squared or absolute deviations from the thermosteric heights. The atmospheric variables (e.g., WSHL, SSHG, SHP, and NAO) are low-pass filtered using a left-sided triangular window with weights decreasing linearly backward from the central point and equal zero forward. The best thermosteric reconstruction is achieved with different WSHL and SSHG|SHP filter window sizes.

The reconstructed thermosteric heights closely approach their targets, especially after 2000 (Fig. 8). The reconstruction captures 96% of the observed variance. The strong correlation (0.98) underscores the robustness of the model in simulating thermosteric contributions to sea level changes over the past three and a half decades, accurately tracking both the overall trend and individual cycles, with observed changes replicated with a 1.0 cm accuracy in 27 out of 33 years (~83%). The overall level of fit that is achieved by using optimally filtered and weighted (scaled) CLS WSHL and SSHG|SHP time series supports our assumptions and demonstrates the effectiveness of the proposed empirical model, making it suitable for further investigation and interpretation of both atmospheric forcing and signal transfer. *This answers the seventh key question.*

The difference between the optimally-fitted WSHL and SSHG|SHP left-side triangularly weighted window sizes of 7 and 13 years, respectively, emphasizes the different roles of the previous winter conditions retained by the water column, known as convective preconditioning[24], and the cumulative effect of summer warming. Indeed, while the winter cooling and mixing are uniquely strong and deep in the Labrador Sea, leaving an immediate trace over a thick layer, the direct effect of summer warming is not that deep. Additionally, anomalous winter cooling situations usually have smaller spatial scales than ocean-wide summer heat waves (*in preparation*). Therefore, it must have taken a longer time and, possibly, a larger region of influence for SSHG to achieve a sizable effect on YASHA and YASLA. By accumulating its signal over a broader domain, SSHG spreads its influence on the thermosteric height over a longer time, hence a longer memory of SSHG changes in YASHA. Yet, while both WSHL and SSHG take turns driving thermosteric height, and the SSHG changes are more influential there on the longer timescales, WSHL dominates in this linkage as a whole. Based on our results, the WSHL–thermosteric interaction is performed through convection, hence the leading role of convective cycles in sea level variability in the CLS domain.

Notably, while the low-pass filtered winter (DJFM) NAO index shows some limited agreement with the yearly averaged thermosteric height, it lacks the precision and detail captured by the presented heat-based model, highlighting the superior predictive accuracy of the heat-based reconstruction approach.

## Linking halosteric height changes to the extreme Arctic sea ice losses

The significant reductions in Arctic sea ice in 2007, 2012, and 2019–2020 (Figs. 8 and 9; and Supplementary Fig. 2) were each followed by pronounced freshening in the upper layer of the Labrador Sea approximately two years later[24] (Fig. 9). This freshening could also be influenced by a recent shift in the Beaufort Gyre's regime from freshwater accumulation to stabilization, with a potential release phase that may contribute to the latest CLS freshening event. In contrast, the Greenland freshwater flux anomaly, which changes more gradually over time, is unlikely to drive rapid freshening events in the CLS[19]. For this reason, we focus our analysis on the impacts of extreme Arctic sea ice losses as the primary driver of intermittent CLS halosteric height changes.

The arrows in Fig. 8 point from the 2007, 2012, and 2020 extreme winter-to-summer Arctic sea ice reductions to the 2009, 2015, and 2022 CLS halosteric height maxima lagged by 2–3 years. The linkages between the extreme points of these two key ocean state variables are inferred from the timing of the respective local salinity minima in the upper (15–100 m) and upper intermediate (300–500 m) layers of the CLS (Fig. 9). The upper layer salinity minima of 2008, 2013, 2022 and 2025, and, lagged by about a year, corresponding upper intermediate layer salinity minima of 2009, 2015, 2023 and 2025, can be traced back to the extreme Arctic sea ice losses of 2007, 2012, 2020 and 2024. The latter are inferred from increased year-to-year Arctic sea ice extent and volume reductions, similarly showing in the time series of August–October means and winter-to-summer reductions of both sea ice metric (Fig. 9). Whenever this happens, an increased quantity of meltwater is passed to the Arctic outflow to be delivered to the Labrador Sea through the Davis Strait (and, possibly later, through the Denmark Strait) to reduce salinity of its upper layer[24].

In contrast to the 2007, 2012, and 2020 extreme releases of Arctic sea ice meltwater being closely followed by the respective CLS halosteric height maxima, the meltwater release of 2024 was trailed by a halosteric height decline. Even though the halosteric height may respond with a rise later on, 2025 stands out as the year when the top 500 m layer of the Labrador Sea experienced the largest (most massive) freshening in the instrumental oceanographic record (Figs. 2 and 9; and Supplementary Figs. 1–4). According to the salinity sections (Supplementary Figs. 3 and 4), this, as well as earlier freshening events, entered the CLS from the Labrador slope side.

The seeming contradiction between the anomalous sea freshening and halosteric height decline between 2014 and 2015 is resolved by considering the broader impacts of the increased Arctic meltwater influx on the full-depth water column. Indeed, through inhibiting winter convection and consequently reducing its depth to 500 m, the sustained freshening of the upper layer led to salinification of the deep intermediate (800–1500 m) layer between 2022 and 2025 (Fig. 9). As winter convection shoaled, the amount of freshwater that reached the deep intermediate layer had reduced, shifting the freshwater balance toward the saltier deep intermediate waters entering the CLS from the neighboring SPNA basins. This resulted in a sustained salinification of the deep intermediate layer, which during 2023–2025 competed with the freshening of the upper and upper intermediate layer over the halosteric height dominance. Obviously, the halosteric height decline seen during 2014–2015 explicitly points to the last-year winner of this everlasting competition.

*Answering our last, eighth, key question, we conclude that a significant reduction in the Arctic sea ice extent may have two counteracting effects on the CLS halosteric height—the first, positive and direct, is achieved through freshening of the upper and upper intermediate layers, while the other, negative and indirect, is brought through inhibiting winter convection resulting in salinification of the deeper layers.*

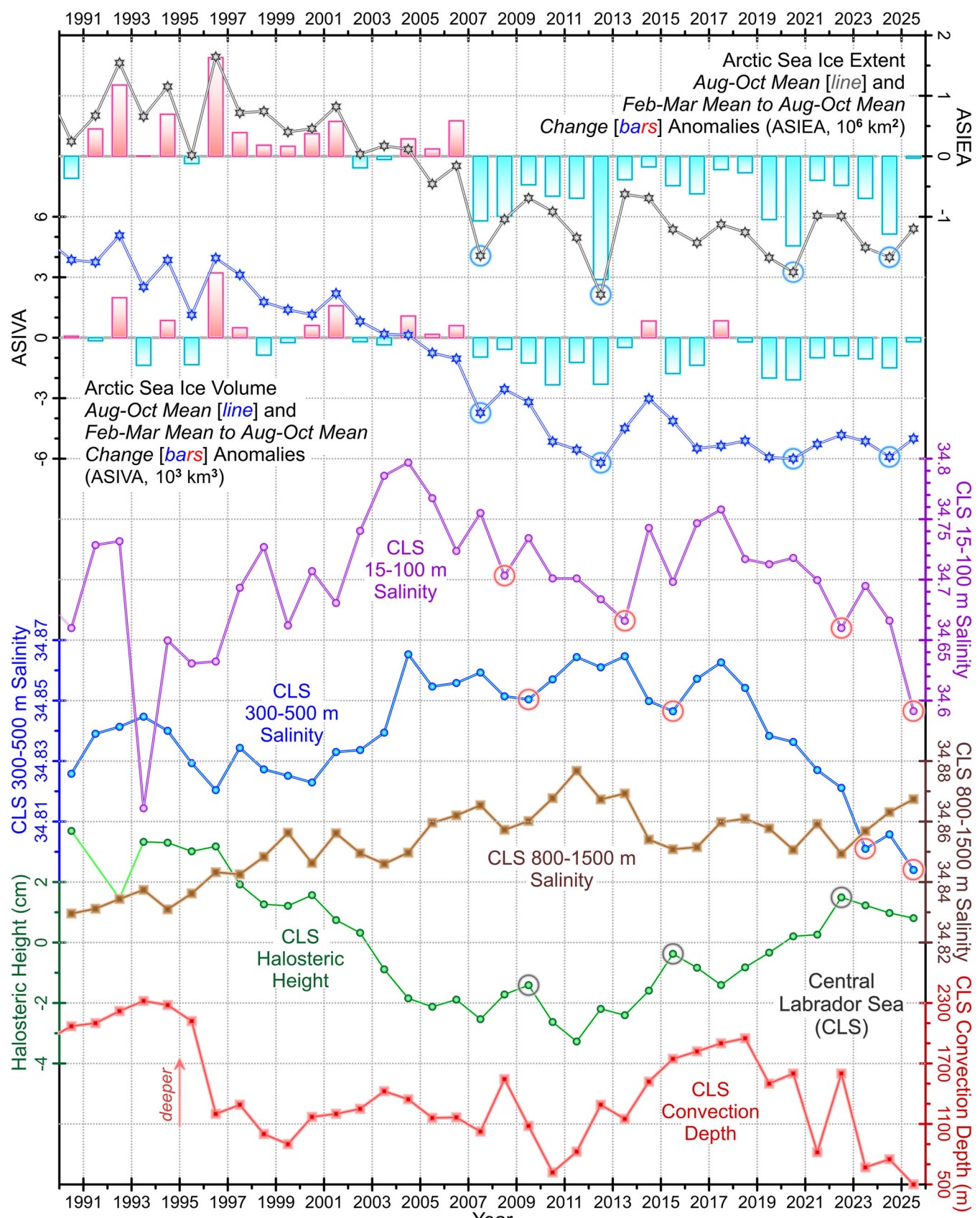

**Fig. 9 | Extreme Labrador Sea freshening and its probable cause.** *Top-down:* The summer (Aug–Oct) Arctic sea ice extent and volume (*lines*), and respective winter-to-summer (Feb–Mar to Aug–Oct) change anomalies (*bars*); de-seasoned, and vertically and yearly averaged, 15–100 m, 300–500 m and 800–1500 m central Labrador Sea (CLS) salinities (*purple, blue, brown*); CLS halosteric height (*green*); and CLS convection depth (*red*). The central Labrador Sea upper (15–100 m) and upper intermediate (300–500 m) layer salinities, and halosteric height likely respond to the extreme Arctic sea ice losses with a two-year lag on average (*circles*).

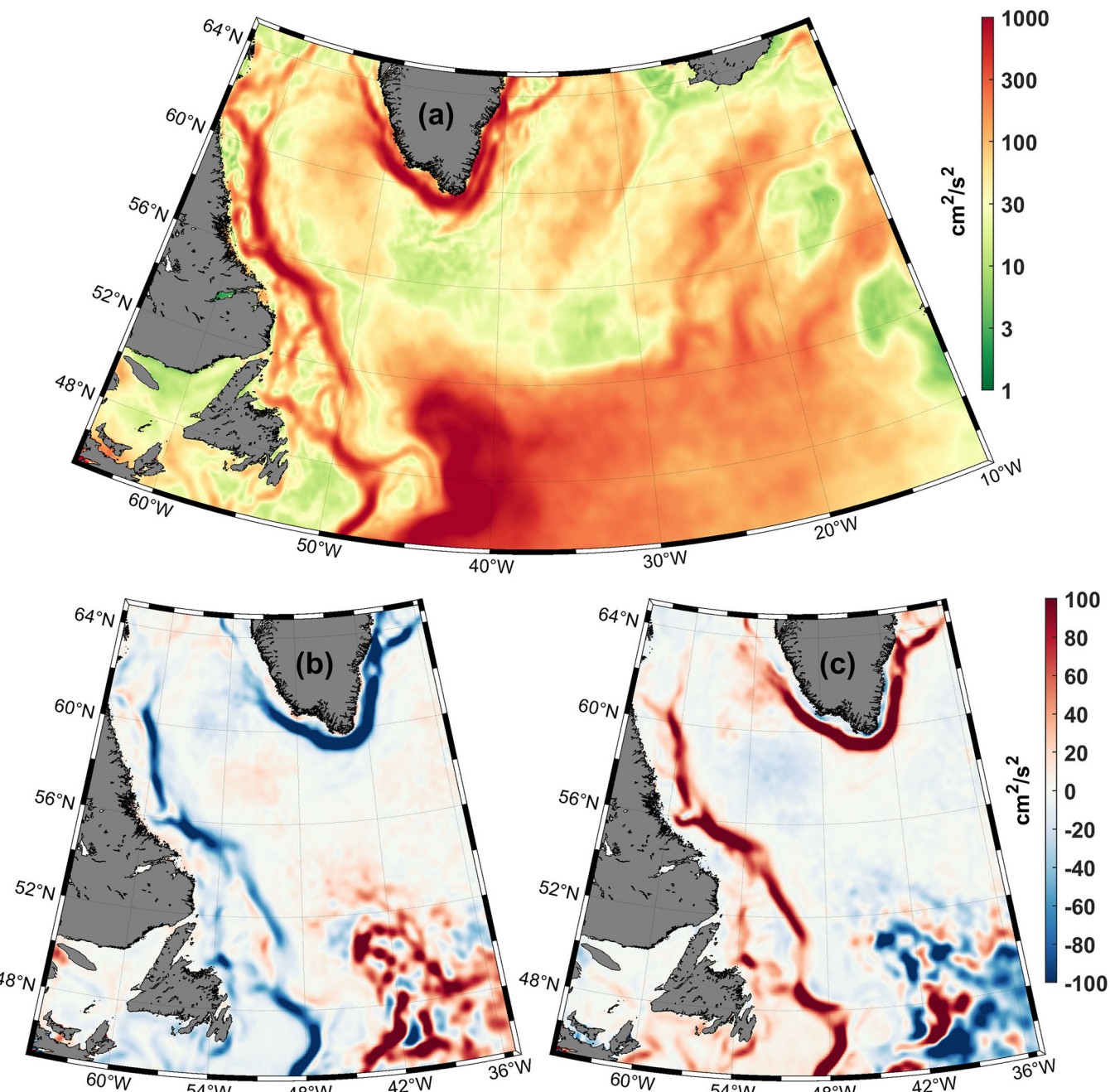

**Fig. 10 | Mean and seasonal kinetic energy (KE) of the Labrador Sea.** Annual mean surface geostrophic kinetic energy (MKE, cm²/s²) derived from the dynamic topography gradients based on the satellite altimetry data and averaged over 1993–2025 (**a**); and the differences between summer (June-August) mean kinetic energy and MKE (**b**), and winter (December-February) mean kinetic energy and MKE (**c**).

However, one might question the lack of substantial annual reductions in Arctic sea ice preceding the 1996–2005 period, during which CLS halosteric height was decreasing alongside rising salinity. This period coincided with exceptionally strong and sustained deep convection from the late 1980s to the mid-1990s, which infused the CLS water column with an estimated 7-meter freshwater equivalent[22,24,43–45]. As the sea entered a convective relaxation phase in 1996, freshwater began to discharge from its intermediate layer into the broader North Atlantic, resulting in a gradual reduction in halosteric height during this relaxation phase.

## Discussion
The Labrador Sea is a unique oceanic basin, where a cyclic pentadal-to-decadal steric height fluctuation can locally oppose, and even reverse,

the global sea level rise, as it happened in 1987–1994 and 2011–2018 (Fig. 1c and 7). These sustained sea level drops were followed by rises, bringing the sea level back to its respective pre-collapse positions. However, in the second case, the level continued to rise even after its full recovery, overcompensating for the preceding drop as recapped and discussed below.

Although, toward the end of the multiyear development of deep convection in the CLS, in 2017, the sea level was more than 1 cm below the 1998–1999's level, it took only eight years for the level to sprout to a record-breaking high. The extreme sea level rise during 2017–2025 was driven by three concurrent processes: (1) the multiyear *de-evolution* of winter convection, which imposed a positive trend on the upper-layer thermosteric height; (2) the covariation, and thus reinforcement, of thermosteric and halosteric height changes in response

to an anomalous freshening of the upper layer; and (3) the emergence of a positive steric trend within the deep layer. These processes allied to enforce a positive trend on YASHA, which ultimately drove the CLS sea level to an unprecedented high.

Unlike the global mean sea level rise, which is dominated by the addition of mass and thermal expansion and shows a negligible halosteric contribution[46,47], the CLS sea level is highly sensitive to a salinity change, and the direction of this change can determine the direction and magnitude of the respective sea level change. Historically, halosteric variability in this region offset thermosteric expansion by nearly a half. However, during the recent sea warming and freshening events, that compensating relationship broke down, and halosteric changes began reinforcing thermosteric ones. This transition marks a fundamental shift in the steric balance of the Labrador Sea, suggesting that future sea level extremes may be increasingly driven by Arctic and Greenland freshwater inputs. As the cryosphere continues to melt, episodes of upper-ocean freshening that amplify thermal expansion could become more frequent. Monitoring freshwater fluxes into the CLS, along with changes in surface forcing and vertical mixing, will be essential to improve projections of regional sea level rise and to assess its broader climatic implications.

### Remaining challenges and future steps

Being the first region, where sea level is shown to be sensitive to the halosteric–thermosteric reinforcement triggered by increased freshwater influx and shoaled deep convection, does not make the Labrador Sea the only unique region, where this effect can accelerate the imminent sea level rise. Therefore, we recommend conducting parallel investigations of climate change and, particularly, polar intensification impacts on sea level in other regions of deep convection affected by planetary freshwater budget changes (e.g., the Greenland and Irminger Seas, and the Sea of Okhotsk in the Northern Hemisphere, and the Weddell and Ross Seas in the Southern Hemisphere).

### Interannual and seasonal water-column mass budget challenges

Although the differences between YASLA and full-depth YASHA are relatively small, they are systematically persistent over time and thus comparable to deep-layer YASHA, particularly since 2016 (Fig. 5). These differences reflect water-column mass changes expressed in equivalent water thickness. The inferred mass changes (black line in Fig. 5) show stable variations over the period of 1993–2005, followed by a continuous decrease between 2005 and 2015, and a subsequent reversal to an increase lasting from 2016 to 2021, amounting to 3 cm and thus becoming a contributing factor to the recent sea level rise and extreme.

The Gravity Recovery and Climate Experiment (GRACE) satellite data offer an alternative method to assess water-column mass changes (Fig. 5). However, the JPL and CSR GRACE mass measurements show inconsistent patterns of interannual changes and opposing unrealistic trends, making any direct comparison with the CLS YASLA–YASHA differences unattainable at the moment. As a promise, detrending of the CSR GRACE time series (green line in Supplementary Fig. 14) improves the alignment of its interannual changes with those of the inferred mass, yet the issue needs further investigation.

Unlike the aforementioned unrealistically divergent JPL and CSR GRACE long-term trends, disagreeing with our assessment of the interannual water-column mass changes (Fig. 5); the regular seasonal cycles extracted from the two GRACE time series, while being nearly identical, are also comparable in magnitude to the seasonal cycle derived from the differences between SLA and full-depth SHA (Fig. 3). The GRACE-based regular seasonal cycles of the CLS water-column mass and the regular cycle derived from the altimetry-based SLA and hydrography-based SHA clearly show a steady mass increase toward summer and a decrease toward winter. Remarkably, these two methods give larger magnitudes of the CLS mass seasonality than the

magnitude of the global mean seasonal cycle, shown for both GRACE products at the bottom of Fig. 3 (*derived by removing low-frequency variability and outliers from individual time series and globally weight-averaging these series basing on areal grid size*). Although there is a time difference between the GRACE and altimetry-hydrography-based CLS seasonal highs, requiring future investigation, the fact that the CLS has a larger magnitude of the seasonal mass cycle than the global mean can be explained by intensification of the cyclonic circulation and hence the divergence of mass over the basin in winter, and weakening of the boundary currents reducing the divergence in summer. This and alternative drivers and mechanisms of the water-column mass changes, including the inverse barometer effect, await a dedicated study supported by both comprehensive observations and numerical ocean models. However, a comparison of the winter and summer mean kinetic energy maps, derived from the dynamic topography gradients based on the satellite altimetry data (Fig. 10) supports our hypothesis with the wintertime intensification of the boundary currents and hence cyclonic circulation act against the convergence of mass over the CLS domain.

Given the overall importance of water-column mass changes to diagnosing and predicting sea level changes, further examinations of the GRACE data quality and derived mass, and finding the reasons behind the unrealistic interannual and longer-term changes in the Labrador Sea are necessary.

## Methods

To assess all components of the sea level budget in the central Labrador Sea (CLS), and fully investigate and understand their changes, we co-analyze:

- the 1992–2025 multi-mission satellite altimetry data;
- the 1948–1989 historical, 1990–2002 high-quality World Ocean Circulation Experiment (WOCE), 2000–2019 Deep-Ocean Observation and Research Synthesis (DOORS), and other ship-based observations;
- the 2002–2025 standard and 2020–2025 Deep Argo float profiles;
- the 2002–2025 satellite data-based gravimetric reconstructions;
- the 1948–2025 atmospheric reanalysis products; and
- the 1979–2025 Arctic sea ice extent and volume compilations[24].

The "*Data Sources*" section offers a brief overview of the respective data sources, while the "*Methods*" section recaps data processing and analysis steps followed in this study.

### Time series decomposition

Our study is based on hydrographic and altimetric observations at specific locations and times without temporal and spatial interpolation, gridding, and smoothing. This approach to data analysis eliminates errors and uncertainties related to interpolation over extensive data gaps, excessive data smoothing, and signal aliasing. Even vertical interpolation of water sample, reversing thermometer, and low-resolution Argo float data is only performed with observations with sufficiently close vertical range of each other. Observations collected within certain geographic locations vary in time with respect to sampling or measurement frequency, contain extensive data omissions or gaps, and therefore form irregular time series. Irregular time series are analyzed by applying a special technique of iterative time series decomposition to all available measurements supplied with their corresponding times[24,48,49]. In this method, each value in a long-term record is regarded as a sum of [1] a regular (i.e., long-term mean or climatological normal) seasonal cycle, [2] irregular seasonal variations, which may be imposed by interannual seasonal phase and amplitude shifts), [3] interannual-to-multidecadal changes and trends, [4] mesoscale and synoptic natural variability (e.g., driven by mesoscale eddies and jet-like currents), [5] high-frequency (e.g., diurnal, inertial, tidal) variability, and [6] instrumental noise, including sampling and

data processing errors. The successive iterations of reevaluation of these components and noise removal are performed until the first three components and residual variance (associated with the mesoscale and higher-frequency components) are stabilized, and no new outliers (errors) are detected and removed.

The regular seasonal cycles of sea level and steric height anomalies (SLA and SHA, respectively), and of thermosteric and halosteric height components of SHA, are shown in Figs. 3 and 4, and Supplementary Figs. 6 and 8. The dots in Fig. 3, and Supplementary Figs. 6 and 8, represent the observed values with removed low-frequency variability. For each analyzed variable, its regular seasonal cycle has been reevaluated on every new (consecutive) iteration of the time series decomposition process− the original data series, after having the outliers and low-frequency variability revealed on the previous iteration removed, is approximated again with a sum of multiple-annual-frequency (i.e., [0, 1, 2, 3, 4, etc.] cycles/year) harmonics, and so on, until a stable outcome of the iterations is achieved. The multiple-annual-frequency cutoff is based on the amount of variance of the original series explained by higher frequencies. The amount of variance explained by the cutoff and higher frequencies combined is negligibly small.

Depending on characteristic features and scales of underlying variability, and temporal changes of sampling frequency and consistency, the low-frequency component is evaluated through either low-pass filtering or polynomial fitting of deviations from the last evaluated regular seasonal cycle. The deviations are either time-bin-averaged or weighted prior to evaluation of the low-frequency component to suppress the effects or biases of temporally uneven data distributions on the time series decomposition. The results shown through the work are obtained with polynomial fitting of the bin-averaged deviation. The markers in Figs. 5−9 and Supplementary Figs. 9−12 and 14 represent yearly averaged deviations from the regular seasonal cycle. The darker red and blue lines in Fig. 5, and darker red, blue, and gray lines in Fig. 6, show the low-frequency components, representing interannual-to-multidecadal sea level and steric height variability. The red, blue, and gray lines in the same figures show the respective low-frequency components with added irregular seasonal variations. The supplementary version of these figures (Supplementary Figs. 9 and 10) has the respective regular seasonal cycles added to the irregular variations.

## Calculation of steric, thermosteric, and halosteric height anomalies

Salinity, temperature, and pressure are used to compute specific volume, enabling the calculation of steric height of a layer of interest (e.g., from bottom to surface), also known as dynamic height[50]. Steric height anomaly and its thermosteric and halosteric components are computed as follows.

Steric Height Anomaly (SHA):

$$\text{SHA} = \int_{p_2}^{p_1} -\frac{V'_{sp}}{g} dp = \int_{p_2}^{p_1} -\frac{V_{sp}(S,T,p) - V_{sp}(S_m, T_m, p)}{g} dp \quad (1)$$

Thermosteric Height Anomaly (TSHA):

$$TSHA = \int_{p_2}^{p_1} -\frac{V_{sp}(S_m, T, p) - V_{sp}(S_m, T_m, p)}{g} dp \quad (2)$$

Halosteric Height Anomaly (HSHA):

$$HSHA = \int_{p_2}^{p_1} -\frac{V_{sp}(S, T_m, p) - V_{sp}(S_m, T_m, p)}{g} dp \quad (3)$$

where $V_{sp}$, $S$ and $T$ are in situ specific volume (i.e., inverse density), salinity and temperature (°C), respectively; $S_m$ and $T_m$ are

climatological mean $S$ and $T$, respectively; $p_1$, $p_2$ are the upper and lower pressure limits of the chosen layers; g is the gravitational acceleration constant as 9.81 m/s². Depending on purpose, the climatological values can be either seasonal (e.g., shown in Fig. 4 and Supplementary Fig. 8, and used to derive respective TSHA and HSHA anomalies shown in Fig. 6) or annual means (e.g., used to derive yearly TSHA and HSHA anomalies shown in Figs. 7−9).

## Reconstruction and prediction of 10−1900 dbar thermosteric height based on optimization of contributions of winter cooling and summer warming

As discussed in the main text, the yearly averaged thermosteric height is reconstructed by optimizing contributions from low-pass filtered total *Winter Surface Heat Loss* (WSHL) and total *Summer Surface Heat Gain* (SSHG) or, alternatively to SSHG, mean *Summer Heat Peak* (SHP), explained in the "Seasonal air-sea heat exchange metrics" subsection. The model employs a filter window, whose weight decreases linearly with each step backward from the central point having the highest weight. The points located ahead of the central point are assigned zero weights. The filter size is defined as the number of points, including the central point, with non-zero weights (one year means retaining unfiltered data). Such a low-pass filter design allows for prorating the contributions of the past forcing conditions to the present ocean state simply and efficiently.

WSHL and SSHG | SHP were low-pass filtered independently from each other with the respective left-side triangularly weighted filter window sizes ranging from 1 to 25 years. The WSHL and SSHG | SHP low-pass filtered series were then added together for all 25×25 = 625 filter size combinations and for each weight of the SSHG | SHP-based contribution selected from a wide range of closely-spaced values. This approach led us to a stable optimal solution of the thermosteric height reconstruction problem. As partially (for four of 25 tested WSHL filter window sizes) shown in Supplementary Fig. 13, the closest match of the observed and reconstructed yearly averaged thermosteric heights is unambiguously achieved with the 7-year and 13-year low-pass filtering of SSHG | SHP, respectively, for a certain weight (∼26) of the SHP relative to WSHL.

Remarkably, the WSHL and SSHG | SHP low-pass filter window sizes, yielding the best agreement with both thermosteric height and 10−1900 dbar ocean heat content, are different. Namely, these sizes are 7 years for WSHL and 13 years for SSHG | SHP. This difference reflects the distinct roles of deep, rapidly progressing (5−7 years) during its active phase, winter convection, and broader, slower-acting interannual variations of summer warming. Our introduced and presented here approach to diagnosing the thermosteric height variations highlights the stronger and immediate impact of winter cooling compared to the longer-lasting cumulative effects of summer warming.

## Statistical analysis

The polynomial, including linear, approximations of the analyzed series are based on the least-squares fitting technique. To improve the stability of higher-order polynomial fits, the input variables (e.g., year) are centered and scaled to the ranges providing the most stable solutions. The 95% confidence interval was derived based on the standard error scaled by a two-tailed Student's t test. The uncertainty of 0.04 cm/year for the global mean sea level trend is cited from ref. 51.

## Data sources

**Along-track satellite altimetry data.** Delayed Mode (DM) and Near-Real Time (NRT) Level-3 1 Hz along-track sea surface height anomalies (SLA), computed relative to a 20-year mean climatology (1993−2012) with ∼7 km (1 Hz) spatial sampling, were used to derive the seasonal

cycle (e.g., Fig. 3 and Supplementary Fig. 6, *lower panel*) and time series (e.g., Figs. 5 and 6, and Supplementary Figs. 7, *lower panel*, 9, 11, and 14) of the CLS sea level. These data were processed by the DUACS multi-mission altimeter system (DM product ID: SEALEVEL_GLO_-PHY_L3_MY_008_062, https://doi.org/10.48670/moi-00146; NRT product ID: SEALEVEL_GLO_PHY_L3_NRT_008_044, https://doi.org/10.48670/moi-00147) and include observations from multiple satellite missions (e.g., ERS-1, ERS-2, Topex/Poseidon, Jason-1/2/3, Envisat, Cryosat-2, Saral/AltiKa, Sentinel-3A/3B, Sentinel-6A, HY-2A/2B, Geosat Follow-On). The dataset was accessed through the Copernicus Marine Service portal (last accessed in September 2025; https://data.marine.copernicus.eu).

**Gridded satellite altimetry data.** DM and NRT Level-4 gridded SLA merges Level-3 along-track measurements from multiple altimeter missions by optimal interpolation (DM product ID: SEALEVEL_GLO_-PHY_L4_MY_008_047, https://doi.org/10.48670/moi-00148; NRT product ID: SEALEVEL_GLO_PHY_L4_NRT_008_046, https://doi.org/10.48670/moi-00149). In addition to SLA, the product provides variables such as Absolute Dynamic Topography and geostrophic currents (both absolute values and anomalies). The gridded dataset, last accessed in September 2025, was used to generate Figs. 1 and 10, and *upper panels* in Supplementary Figs. 6 and Fig. 7. A global sea level time series derived from along-track data of multiple altimeter missions (Fig. 1c), last accessed in September 2025 to validate the gridded global sea level time series, is provided by the NOAA Laboratory for Satellite Altimetry (https://www.star.nesdis.noaa.gov/socd/lsa/SeaLevelRise/LSA_SLR_timeseries.php). Both global mean sea level time series were Glacial Isostatic Adjustment (GIA) corrected by +0.3 mm per year[15,46]. Since GIA correction is not part of the standard satellite altimetry processing and no publicly available model provides a consistent correction for our regional analysis, we do not apply GIA correction in our trend and uncertainty estimates.

**Satellite gravimetry data.** To evaluate mass contributions to sea-level change, we use two GRACE/GRACE-FO datasets, including the JPL GRACE and GRACE-FO Mascon Ocean, Ice, and Hydrology Equivalent Water Height Coastal Resolution Improvement (CRI) Filtered Release 06 Version 02 (TELLUS_GRAC-GRFO_MASCON_CRI_GRID_RL06_V2)[52–54] and the CSR GRACE/GRACE-FO RL06 Mascon Solutions (Version 02)[55,56]. The JPL dataset, processed and distributed by the Jet Propulsion Laboratory (Data Portal | Data – GRACE Tellus), provides monthly equivalent water thickness anomalies (in cm) at its nominal resolution is 0.5° × 0.5°. The CSR dataset, processed by the Center for Space Research and accessible at http://www2.csr.utexas.edu/grace, offers monthly 1/4° × 1/4° global coverage. The effective resolution[52–54,56] of both monthly gravity fields is 3° × 3°. Spanning January 1993 to present, GRACE datasets include data gaps due to technical issues, with details available at the GRACE mission portal (https://grace.jpl.nasa.gov/data/grace_months/). Both satellite gravimetry datasets were last accessed in September 2025.

**Multiplatform hydrographic measurements.** This study integrates hydrographic data from multiple sources, including profiling Argo float (2002–2025), historical water sample and reversing thermometer ship-based (1948–1985), and recent high-resolution ship-based observations, to construct the CLS hydrographic time series (Figs. 2, 4, and 9), and steric, thermosteric, and halosteric height time series (Figs. 3 and 5–9).

Comprehensive details on hydrographic data quality control, editing, and merging procedures are provided in ref. 24, while here we recap the history of these observations. Systematic observations in the Labrador Sea date back to the late 1940s, with the major contributions for more than two decades being associated with the International Ice Patrol, U.S. Coast Guard, and Ocean Weather Ship Bravo[24,49,57,58]. Dedicated research missions, such as the 1966 and 1976 CSS Hudson expeditions, raised attention to the Labrador Sea as a key intermediate-depth water source of the North Atlantic.

The Atlantic Repeat Hydrography Line 7-West (AR7W) line surveys conducted by the Bedford Institute of Oceanography over the period of 1990–2019[24,25,58,59] had provided, exclusively for this period, measurements of exceptionally high accuracy. The extensive high-quality Labrador Sea hydrographic profiling and sampling initiative (Supplementary Table), carried as part and in support of the World Ocean Circulation Experiment (WOCE, 1990–2002) and the Deep-Ocean Observation and Research Synthesis (DOORS, 2000–2019), and terminated after 2019, provided high-resolution full-depth temperature and salinity measurements with the respective accuracies of 0.001 °C and 0.0015, or better, for the period of 1995–2019, and 0.002 °C and 0.002 for the period of 1990–1994.

The Argo float profiles, massively increasing in numbers since 2002, reduce our reliance on ship-based observations, offering year-round 0–2000 m and full-depth data to resolve seasonal and inter-annual variability, particularly in years without ship surveys (e.g., 2017, 2021). Rigorous quality control, including calibration of ship-based sensors and advanced Argo float data quality control, validation and correction carried forward from our previous study[24,25,58,59], ensures consistency and accuracy across datasets. Advanced, adapted to routine utilization of observations from various platforms (e.g., floats, ships), data processing techniques further improve temporal and spatial resolution, enabling detailed analysis of long-term trends and seasonal cycles in the CLS.

**North Atlantic Oscillation (NAO).** The North Atlantic Oscillation (NAO) is a key teleconnection pattern affecting atmospheric conditions in the Labrador Sea[24,60]. A positive NAO phase increases the sea level pressure (SLP) difference between the Icelandic low and Azores high, intensifying westerlies that bring cold, dry air to the region. Conversely, a negative NAO weakens westerlies, leading to warmer conditions. The winter NAO index (Fig. 8 and Supplementary Fig. 2) is derived from December-to-March principal component-based values using the first empirical orthogonal function of 500-mbar height anomalies (https://www.cpc.ncep.noaa.gov/products/precip/CWlink/pna/nao.shtml).

**Seasonal air-sea heat exchange metrics.** The atmospheric variables and reconstructed *Winter Surface Heat Loss* (WSHL), *Summer Surface Heat Gain* (SSHG) and *Summer Heat Peak* (SHP) used in this study (Fig. 8 and Supplementary Figs. 2 and 13) are based on the NCEP/NCAR Reanalysis datasets[61,62] (https://psl.noaa.gov/data/gridded/data.ncep.reanalysis.html; https://psl.noaa.gov/data/gridded/data.ncep.reanalysis2.html), provided by NOAA, USA. The Reanalysis products (R1 and R2) of the highest available resolution (6-hourly) were compared and jointly utilized to achieve comprehensive and detailed atmospheric data coverage.

*Net Surface Heat Flux* (NSHF) values were computed by combining the shortwave and longwave radiative fluxes and turbulent latent and sensible heat fluxes from 6-hourly NCEP/NCAR fields, averaged over the central Labrador Sea (CLS, Fig. 1, red contour). The start and end points of an individual winter season were defined from NSHF sign reversals. Starting in late fall with a positive-to-negative NSHF transition, the cooling or winter season ends in early spring, when NSHF changes from negative to positive. Summer seasons were defined as the periods between the spring and fall NSHF sign reversals, when the net surface heat flux remained consistently positive, indicating ocean heat gain. WSHL was determined by integrating NSHF over a full cooling period, excluding short-term reversals, which have a negligible impact on the total heat loss.

Integrating NSHF over a warming period gives SSHG, averaging it over a period (e.g., 31-day long) centered on an outgoing flux high gives SHP.

**Arctic sea ice extent and volume.** Daily Arctic sea ice extent and Arctic sea ice extent volume[63] data were downloaded from the National Snow and Ice Data Center (https://nsidc.org/home) and Polar Science Center (https://psc.apl.uw.edu/research/projects/arctic-sea-ice-volume-anomaly/), respectively. Annual winter-to-summer Arctic sea ice extent reductions (Figs. 8 and 9; Supplementary Fig. 2) were calculated by interpolating small data gaps (10 days), computing 1979-2025 daily means from gap-free years, and subtracting these means from daily values. Late-winter (Feb–Mar) and late-summer (Aug–Oct) averages and their differences were derived from the anomalies.

## Data availability
The extensive data archives compiled for this study are continuously revised and updated. All related information, fully updated data and results are available from the corresponding author (Igor Yashayaev, emails: Labrador.Sea@gmail.com; Igor.Yashayaev@dfo-mpo.gc.ca).

## Code availability
MATLAB, Visual Basic for Applications (VBA), Golden Software Surfer and Grapher, and MATLAB M_Map toobox[64] were used for computations and visualization. Continuously updated codes, and related instructions and guidance are available from the corresponding author (Igor Yashayaev, emails: Labrador.Sea@gmail.com; Igor.Yashayaev@dfo-mpo.gc.ca) upon request.

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

## Acknowledgements

The authors thank Joël Chassé (Maurice Lamontagne Institute), Guoqi Han (Institute of Ocean Sciences), and Don P. Chambers (College of Marine Science, University of South Florida) for their critical, insightful, and constructive suggestions, helping to improve this paper.

## Author contributions

I.Y. assembled, quality-controlled, calibrated, processed, and integrated the ship-based and Argo float hydrographic profiles; performed time series analyses, including decomposition of sea level and steric height variables into components (e.g., seasonal, low- and high-frequency, thermosteric and halosteric); processed atmospheric and Arctic sea ice extent and volume data; and conducted a series of experiments on reconstructing thermosteric and halosteric height changes from the atmospheric and Arctic sea ice data. Y.Z. Downloaded and converted to Matlab mat files along-track and gridded, delayed mode, and near-real time satellite altimetry, and GRACE gravimetry data; analyzed spatial distributions of the mean sea level topography, sea level rise rate, and kinetic energy; and conducted a literature review. I.Y. created Figs. 2–9 and all Supplementary Figures. Y.Z. Created Fig. 10. Both authors contributed to making Fig. 1. Both authors participated in conceptualizing the work, analyzing and interpreting the results, discussing and writing the manuscript.

## Competing interests

The authors declare no competing interests.
