## [Transparent Peer Review file · Nature Communications]

Concurrent warming, freshening and cessation of deep convection in the Labrador Sea raised its sea level to a record high

Corresponding Author: Dr Igor Yashayaev

Version 0:

Reviewer comments:

Reviewer #1

(Remarks to the Author)

The authors utilize data obtained through various observation methods to thoroughly elucidate the factors influencing sea surface height variations in the Central Labrador Sea (CLS). The novelty of this study lies in three aspects: (1) demonstrating the strong covariability between sea level anomaly (SLA) and steric height anomaly (SHA) in CLS, (2) separating and discussing the contributions of temperature (heat) and salinity (freshwater) to SHA variability, and (3) presenting results from 2023 onward on the relationship among temperature, salinity, and deep convection intensity. These findings are highly advanced in the fields of ocean physics and climate dynamics and are valuable for publication.

However, the current draft contains too much additional content, making it difficult to grasp the key points. There is too much description in the results section about previous findings obtained by researchers, including the authors themselves. Since the purpose of Nature Communications is not to publish review articles, not all but a significant portion of these descriptions should be moved to the introduction. For example, the discussion of previous findings in lines 141–153 should be either relocated to the introduction or omitted. However, the issues mentioned do not negate the scientific value of the research and are not fundamental problems.

I will point out one issue in the interpretation of the results obtained in this study that I believe is problematic. We have confirmed a strong long-term correlation between SLA and SHA. However, since SLA changes slightly earlier, I doubt the validity of the logic that SHA changes cause SLA changes. At the very least, a discussion should be added regarding the possible reasons why SLA appears to be leading.

Except for the above-mentioned points, most of the sentences are written logically and accurately, and the figures are also clearly presented, making the manuscript well-structured for publication as a research paper. If the suggested restructuring of the manuscript does not require significant time, I suggest this draft qualifies as a minor revision.

Reviewer #2

(Remarks to the Author)

The paper uses long-term observations from Argo, satellite altimetry, and other sources to show the rise (and occasional fall) of sea level in the central Labrador Sea since the 1990s. It shows that local thermal expansion in the top 2 kilometers accounts for most of the rise, with salinity effects counteracting thermal expansion until the last decade, with smaller effects due to changing mass in the water column and to steric effects below two kilometers. The paper also traces thermohaline variability back to the 1950s and analyzes seasonal influences on temperature and salinity.

The methodology seems sound. The thorough analysis of sea level changes is very interesting. The paper does not make a strong case for the significance of central Labrador Sea sea level. I believe that this is a worthwhile topic because of the importance of the Labrador Sea for its role in AMOC and for the interesting currents and water mass formation there. Paper references the importance of sea level for the [rather small] human populations along the Labrador and Greenland coasts, though the relationship is somewhat indirect given the complex spatial pattern of sea level rise [Figure 1]. This stuff seems very interesting to physical oceanography (my field) and somewhat interesting to anyone interested in sea level rise; not sure of how much interest it would be for others in climate.

I found the paper a little difficult to read in places. Perhaps more focus on main points and less on various observational details?

Suggestions for Improvement

1. Organization of results. I found it confusing that some of the key results about steric height are in the Intro rather than in the results section. These seem to be a new result, but since they are in the Intro, this was not so clear. Similarly, paragraph 2 of Discussion also belongs in the Results section. The presentation of results in Intro and Results sections alternates between annual average and seasonal cycle in a way that I found hard to follow. I suggest starting with annual average results for sea level and steric heights, then discussing seasonal cycle, then annual average sea level going back further in time, then influence of seasonal cycle in heat flux (2nd subsection of discussion).

2. Interpretation of results. Global sea level rise is a key feature of global warming. Paper needs to include more context from global sea level rise. For instance, Guerou et al (2023) show GMSL 10-year trends of about 2.5 cm/decade in 2000 and 4.4 cm/dec in 2015, which gives a reference point for what constitutes a “fast” sea level rise. To me it was very interesting how the SLR signal was dominated by steric height, given that globally, changes in steric height seems to account for less than half of the change in sea level (Dieng et al, 2017). Other important factors include gravitational effects associated with geographical redistribution of mass, and changing circulation. I would think that there is some change in subpolar gyre strength which would change the sea surface height gradient between the edge and center of the gyre and hence change sea surface height in center of gyre. Finally, Labrador Sea is close to the “warming hole” of surface warming. When I look at 11 yr running mean (55-60 W, 55-60 N) from Berkeley Earth, I see a big jump in SST in the 1930s, a slow decline till around 1990, big jump over next 15 years, then slow decline again.

Seems like this is all worthy of discussion and it’s possible that the dominance of steric effects is more surprising – hence interesting – than paper lets on.

Dieng, H. B., A. Cazenave, B. Meyssignac, and M. Ablain (2017), New estimate of the current rate of sea level rise from a sea level budget approach, *Geophys. Res. Lett.*, 44, 3744–3751, doi:10.1002/2017GL073308.

Guerou et al, 2023: Current observed global mean sea level rise and acceleration estimated from satellite altimetry and the associated measurement uncertainty, *Ocean Sci*, 19, 431-451, <https://os.copernicus.org/articles/19/431/2023/>.

3. Significance of Record Highs. This point is closely related to #2. The paper highlights how the reversal of halosteric effects makes 2024 SLA slightly above previous records rather than slightly below. Does this small difference matter? A bigger point, which paper does not say much about, is that variability on about 5 year timescales is strong enough to temporarily reverse the long-term upward trend.

Some Details

Fig 1

* Where does velocity and sea level data come from?

* Why show trends in highs and lows (bottom panels)?

* Showing the trends makes me want paper to say more about them. It looks like the surface geostrophic current around the basin is not getting stronger, though the Lab Current is. It’s a little distracting, since rest of paper focuses on CLS temporal behavior.

Fig 2 and some subsequent figures. Make separate figures to show annual cycle and interannual variability. For annual cycle and monthly (weekly?) data points, select a subset (10 years?) and superimpose fitted curves shown in current version of figure. In separate figure, show low-frequency data as before, but with altimetry and 10-3300 dbar hydrography superimposed and either no annual cycle or annual cycle replaced by one curve connecting all the annual maxima of fitted curve and another curve connecting annual mins. Most of the discussion of these figures is about interannual variability and comparisons between different quantities (height, thermosteric component, etc) for which the seasonal cycle is a distraction.

Lines 139-176 & Fig 4. Very hard to read. First sentence is painful. It is followed by description of Labrador Sea budgets which maybe should go in Intro when motivating the analysis of changes in steric heights. It’s hard to see the breaks in deep convection from Fig 4. If deep convection is signaled by deep annual maximum in bottom of mixed layer, then temperature, salinity, and density fields all look to me like they are showing different patterns. It’s also hard to detect these breaks given aspect ratio of figure. I think the bottom panel is showing how deep one must go for density to be 0.005 kg/m³ denser than surface, but I’m not sure. It’s confusing that shut-downs of deep convection are listed in first paragraph, but then how one monitors deep convection from property fields is described in the middle of second paragraph.

Equations 1-3. Simplify notation such as removing subscripts from S_A, V_{sp}, T_C and using subscript m rather than mean to denote mean. Typo line 66 “equations xx”.

Lots of typos: Please do some additional proofreading. Many instances of “heigh” instead of “height”, subsection title “Where there other occurrences...” instead of “Were there other occurrences...”

Long subsection titles. Paper subsection titles make actual statement of fact, which is useful, but long section titles are

harder to read.

Winter convection regime changes as the primary driver of sea level variability in the Labrador Sea  Winter convection regime changes cause sea level variability

A reversal in the upper layer temperature-salinity steric balance drove the sea level to a record high  Reversal of temperature-salinity relation drove record high sea level

Were there other occurrences of halosteric reinforcements of thermosteric height changes in the Labrador Sea in the past? 

Pre-1990 halosteric reinforcements of thermosteric height changes

Version 1:

Reviewer comments:

Reviewer #1

(Remarks to the Author)

The authors responded sincerely to the reviewers' comments. I have confirmed in the Discussion that, regarding my concern about the timing mismatch in the temporal variations of SHA and SLA, it is not easy to demonstrate this with the current observational data, and that, in addition to changes in basin-scale circulation, other possible causes for this mismatch are also mentioned. I believe that the manuscript is suitable for publication as a paper in Nature Communications.

(Remarks on code availability)

Reviewer #2

(Remarks to the Author)

This is a review of the revised manuscript. The revision is better organized and clearer. It tells an interesting story of rapid sea level rise in the Labrador Sea driven by thermal expansion and augmented by a recent change in salinity effects, both of which the authors related to atmospheric forcing. I recommend publication.

I have a few remaining suggestions which I leave to the authors' discretion; I do not need to review the manuscript again.

1) Correspondence between numbered questions in Key Research Questions and subsections in Results. The Results subsections partly correspond to the questions. Perhaps it would be helpful for readers if each question corresponding to a results section. Questions 3-5 all pertain to halosteric effects and could be combined into a single item. The last couple of sections raise and answer questions that are not mentioned in the original listing.

2) Figure 1ab. For filled contour maps, in general I recommend using a small number of discrete shades/colors, rather than a continuous or near-continuous color palette. This makes it easier to see the range of values present in a field. Currently it is hard to say much more than that there is a lower (green) and higher (purple) part of the gyre in 1a and a rapidly (purple) and less rapidly (white) rising sea surface in 1b. If the authors want to show any gradations beyond that, a few discrete shades would be helpful.

(Remarks on code availability)

Igor Yashayaev
1 Challenger Drive, Dartmouth
Dartmouth, NS, B2Y 4A2, Canada
Igor.Yashayaev@dfo-mpo.gc.ca
(782) – 640 - 9437

August 1, 2025

Responses to the Reviewers of “*Concurrent Warming and Freshening Led to a Record-High Sea Level in the Labrador Sea*” by Igor Yashayaev and Yang Zhang

Dear Reviewers,

We immensely appreciate your attentive, thorough, constructive and positive handling of our submission. The reviews contained invaluable suggestions that allowed us to improve our paper, making it more outreaching and visible to academic community, policymakers and general public. All your suggestions, questions and requests have been addressed in our point-by-point response below. To streamline the responses and ensure that we have not missed any, we split both reviews in individual points, and merged, grouped and ordered those by topics. We also highlight the recommendations that, in our opinion, have strengthened the prospective impact of our paper, by, for example, comparing the rates of the recent sea level rise in the Labrador Sea, subpolar North Atlantic and World Ocean.

We also agree that the articles submitted to *Nature Communications* are expected to be narrower in scope and shorter than the work you so generously and helpfully reviewed. Unfortunately, the reported sea level rise to an extreme is convincingly a result of five different processes, all of which has recently acted in the same way. Explaining the complexity of the studied interactions in a plain language does not also help to shorten the story. Please also take into consideration the diverse content of the analyzed data. We hope that you agree with our justification of retaining a broader-than-expected scope of work. Furthermore, an in-depth analysis of the mechanisms behind the reported changes seemed appropriate for the Climate Extreme issue of *Nature Communications*.

Even though the paper is focused on the recent sea level rise in the subpolar North Atlantic, the anomalous trend we see there could not be explained without showing a detailed fully up-to-date temperature, salinity and density records. These records, just by themselves, point at recent and still ongoing extreme ocean state conditions, underlining our subsequent sea level analysis. Your recommendations helped us to restructure the storyline, hopefully making it clearer and easier to follow. In particular, the mentioned oceanographic records are now found in the “*Key Research Questions*” section. We think that raising most of the key questions just by looking at these series adds a spin to the whole story. Each question is referenced in the “*Results*” section whenever an answer is given. Furthermore, the “*Supplementary Information*” section has massively grown in size, extending the oceanographic data synthesis, showing alternative versions of the time series figures, etc.

In the table below, we numbered all points of the reviews, color-coded by source, and provide our responses to each. Additionally, we identified a few key points as part of our general guidance and revision undertaking. The first group of points concerns the structure, the second – general scientific issues, and the third – edits and minor (paragraph and subsection) restructuring.

#	Issue, concern, request, comment, suggestion	Response, handling
General Instructions		
1	Authors: Address all reviewers' concerns without exception.	We provide a point-by-point response in this table, with all the raised suggestions, recommendations and concerns sorted by subjects and originator.
2	Authors: Make sure all changes are reflected in the manuscript text file with track changes.	Because of major restructuring of the paper (changing order of entire sections and paragraphs), the tracked changes might be hard to follow. Nevertheless, we provide the manuscript text file with tracked changes.
3	Authors: Provide here a thorough convincing in-depth response to each request (if any) unaddressed in the revised manuscript.	All requests are excellent and helpful to the authors. One comment might involve our partial misunderstanding, and one was of discussive nature – both have been fully addressed and thoroughly analyzed below.
4	Reviewer #2: I found the paper a little difficult to read in places. Perhaps more focus on main points and less on various observational details?	We followed this recommendation as part of restructuring and editing of the entire manuscript and individual sentences. All answers to the key questions are now explicitly marked, which hopefully helps.
Overall Assessment		
5	Reviewer #1: The authors utilize data obtained through various observation methods to thoroughly elucidate the factors influencing sea surface height variations in the Central Labrador Sea (CLS).	We thank the Reviewer for identifying the main purpose of our work.
6	Reviewer #1: The novelty of this study lies in three aspects: (1) demonstrating the strong covariability between sea level anomaly (SLA) and steric height anomaly (SHA) in CLS,	We thank the Reviewer for this positive insightful recap of the novelty of our study. As part of our work on this revision, we have extended the time series through April of 2025, emphasizing the main statements, conclusions and interpretations. We will pursue updating and improving the results. The final version (if we will

	(2) separating and discussing the contributions of temperature (heat) and salinity (freshwater) to SHA variability, and (3) presenting results from 2023 onward on the relationship among temperature, salinity, and deep convection intensity. These findings are highly advanced in the fields of ocean physics and climate dynamics and are valuable for publication.	be fortunate to take it that far) will all series updated after all text revisions are done. Having the longest records will add uniqueness to our study.
7	Reviewer #2: The paper uses long-term observations from Argo, satellite altimetry, and other sources to show the rise (and occasional fall) of sea level in the central Labrador Sea since the 1990s.	We thank the Reviewer for summarizing the scope of our work.
8	Reviewer #2: It shows that local thermal expansion in the top 2 kilometers accounts for most of the rise, with salinity effects counteracting thermal expansion until the last decade, with smaller effects due to changing mass in the water column and to steric effects below two kilometer. The paper also traces thermosteric variability back to the 1950s and analyzes seasonal influences on temperature and salinity.	We thank the Reviewer for summarizing the main results, including the seasonal influences.
9	Reviewer #2: The methodology seems sound. The thorough analysis of sea level changes is very interesting.	We appreciate this assessment of the technical aspects and thoroughness of the study.
10	Reviewer #1: However, the issues mentioned do not negate the scientific value of the research and are not fundamental problems.	We truly appreciate the positivism and constructivism of this statement! However, we think that all of the Reviewer’s suggestions that we followed in detail are vitally critical for the scientific value of the paper. Important results would be likely left unnoticed and ignored, if not properly organized and presented. Any paper linking different small and big subject areas is not easy to write and read, even harder to review as requirements are higher to connectivity studies. So once again, our great appreciation to the team on the task.

		We would definitely be keen to hear if the present version has all of the raised concerns fully addressed.
11	Authors: Revisit, and deepen all our result interpretations.	Naming just a few steps undertaken to deepen the interpretations of our results:  (1) By extending the high-resolution time series to April of 2025, and annual metrics to 2024 and 2025, we increased the separation between the extreme ocean states (freshwater content and sea level) observed presently and in the past; (2) Also as suggested by Reviewers, we added explicit comparisons of sea level changes in the Labrador Sea, subpolar North Atlantic (except the Labrador Sea), and World Ocean. This allowed us to highlight the uniqueness of the Labrador Sea in view of the ocean-wide and global sea level changes; (3) We strengthened our interpretation of the faster than ocean-wide and global, sea level rise in the Labrador Sea to record high as a result of coaction of multiple factors, as also mentioned by Reviewer #2 (Row #8).
Structure of the manuscript		
12	Authors: Make a clearer separation between the two groups of the results – supportive (temperature, salinity, density) and core (sea level and its components), and reorganize the paper focusing on our key points related to sea level.	Clearly identifying and addressing key points has always been a challenge for complex climate systems, and this time we tried doing it better than in the initial submission. Both Reviewers seem to agree that the key points are fairly strong. There are two groups of new, equally important results presented in the study: (1) the recent changes in the seawater properties and their causes, and (2) variability of sea level and steric height, and their drivers. Even though, our main focus is on the second group, we would not be able to explain sea level changes without showing and discussing the seawater property changes. The reviewers rightly noted that we have results scattered between the “Introduction” and “Results” sections, which makes it hard to identify the key ones, i.e., those from the second group.

		In the present revision of the manuscript, we solve this issue by splitting the “Introduction” section into two subsections – “Background” and “Key Research Questions”. We show all results about water property changes in the second subsection, which can still be regarded as a collection of novel results, laying the basis for the steric height and hence sea level analysis, yet separated from the key results and from the overview of previous works. We hope that this arrangement will not be rejected as it really helps to separate critically important supportive and key points, and given the importance of the region, backed by both reviewers, accommodate multiple interconnected discoveries (e.g., record-shallow convection, sustained freshening caused by Arctic sea ice melt and extreme sea level) in Nature Communications.
13	Reviewer #1: However, the current draft contains too much additional content, making it difficult to grasp the key points.	We agree with the Reviewer. The paper is indeed heavily loaded with information concerning the recent and ongoing changes in the Labrador Sea environment. Our toughest change was to co-accommodate the sea level and supporting observations. The present state of the Labrador Sea is really unique, which we needed to emphasize as it adds to the key point of the paper, that is the sea level rise. Given the fact that the observations reveal exceptional changes in all analyzed fields, splitting the paper in two would break the important linkages in the system we tagged as our key points. Instead, we now provide a solution by separating basic hydrography from sea level as explained in Row #12.
14	Reviewer #1: There is too much description in the results section about previous findings obtained by researchers, including the authors themselves. Since the purpose of Nature Communications is not to publish review articles, not all but a significant portion of these descriptions should be moved to the introduction. For example, the discussion of previous	We thank the Reviewer for explaining the general concern presented in Row #13. As a matter of fact, the recommendation to relocate the discussion of the previous findings motivated us to restructure the results as explained in Rows #12 and #13. The discussion of the previous findings is turned into an update of those, which would be worth presenting in a standalone paper, but it is also critical for the interpretation of the imminent sea level rise (the Arctic is still

	findings in lines 141–153 should be either relocated to the introduction or omitted.	melting, Labrador Sea convection shoaling), so we worked around the raised concern by introducing a standalone “Key Research Questions” subsection. Oppositely to this recommendation, Reviewer #2, in Row #15, suggested to relocate some parts of “Introduction” to “Results”. The two suggestions, which seem to be in conflict with each other, but in reality are not, made a perfect sense to us. The two excellent suggestions (Rows #14 and #15) motivated us to flip the story flow a bit – instead of showing an extreme sea level in the “Introduction” section, and introducing and discussing high-resolution hydrography in the “Results”, we now fully retain the sea level extreme story until for “Results”, and use hydrography as part of motivated questioning,
15	Reviewer #2: Suggestions for Improvement, 1. Organization of results. I found it confusing that some of the key results about steric height are in the Intro rather than in the results section. These seem to be a new result, but since they are in the Intro, this was not so clear.	As we noted in Row #14, both of our Reviewers advise to relocate parts of the story, but in opposite directions. This is absolutely no controversy here, because the reviewers meant different elements of Introduction and Results. The suggestion in this line likely points to the extreme sea level trend that came as a motivational point in the first submission. Thanks to both our Reviewers, we now motivate the study by showing the most recent temperature and salinity changes, and challenging ourselves with their roles in sea level variability. However, while moving our main results based on our preferred multi-mission along-track data into Result, we added a panel to Figure 1 to provide another motivation point – the regional uniqueness of the Labrador Sea with respect to sea level variability. We were actually debating if this panel should be turned into a stand-alone figure in Discussion, but decided to make it as present. It does not steal the thunder from Figure 5 that is showing the agreement between dynamic and steric heights.
Summarizing our responses in Rows #14 and #15: We would greatly appreciate to hear the Reviewers’ opinions on the undertaken reorganizations, which resembles “castling” in chess, although we refrain from asking ourselves which of the two is the king, and which is a rock here.		

16	Reviewer #2: Suggestions for Improvement, 1. Organization of results. Similarly, paragraph 2 of Discussion also belongs in the Results section.	The Reviewer refers to the “Reconstruction of the interannual thermosteric height changes” subsection, which we should keep it in the same section with the next subsection on the halosteric changes. This is a very good suggestion, because we originally debated whether our sea-level (or rather its leading thermosteric component) simulation technique that gave an unexpectedly strong result, should be placed with results or conclusions. Honestly speaking, the main reason to give it to the “Discussion” was to unload the “Results” and give something to the “Discussion” section. Presently, thanks again to the Reviewers, we added a comparison of the central Labrador Sea, subpolar North Atlantic and global sea level change, which is just right for the “Discussion” section. We think this makes more sense than what we had before, adding another credit to the Reviewers.
17	Reviewer #1: Except for the above-mentioned points, most of the sentences are written logically and accurately, and the figures are also clearly presented, making the manuscript well-structured for publication as a research paper.	Given that English is not a first language for both authors, we regard this statement as a high praise of the effort given to the writing. Thanks! We tried to improve the overall layout and the text, and will greatly appreciate further guidance.
18	Reviewer #1: If the suggested restructuring of the manuscript does not require significant time, I suggest this draft qualifies as a minor revision.	We thank the Reviewer for their supportive and constructive attitude. Regardless of the time required to address all raised concerns and follow all suggestions, we have voluntarily undertaken a thorough major revision of the whole work updating and improving most of the results and graphics, comparing with other domains, etc. We highly enjoyed the interactive revision process, and will appreciate critical suggestions and recommendations on the revised manuscript.
19	Reviewer #2: The presentation of results in Intro and Results sections alternates between annual average and seasonal cycle in a way that I found hard to follow. I suggest starting with annual average results for sea level and steric heights,	We thank the Reviewer for this recommendation. While working on the initial version, we kept changing our minds on the order of the seasonal and interannual segments. At the end, we decided to start with the interannual signals capturing one of the key messages. However, we totally agree

then discussing seasonal cycle, then annual average sea level going back further in time, then influence of seasonal cycle in heat flux (2nd subsection of discussion).	with the Reviewer that jumping back and forth between the seasonal cycles and interannual changes, regardless of our initial motivation, is somewhat illogical. We now reverted to our very original plan, where instead of showing an extreme signal first, we progress from shorter to longer time scales. We start with regular seasonal cycles in all variables and at all depth levels (that have sufficient seasonal coverage), and explain in the text why the regular or climatological seasonal cycles are needed at first place. Even though the regular seasonality is a relatively dull topic, showing high-resolution seasonal cycles for related variables gives us, among other things, of the total range of variations in combination with interannual changes. For a sceptical reader, we start showing results by giving an idea how well the sought climatologies are defined. One of the other advantages of starting with the seasonality is that it allows us to increment the level of complexity as we proceed with the other figures. Also, since we are now showing de-seasoned interannual series, as the Reviewer again very rightly suggested, it is important to show a curious reader what the background seasonal cycles really are. We then proceed following the Reviewer's suggestion and basically as we had in the initial version, except for the seasonal cycles, of course. Regarding “then influence of seasonal cycle in heat flux”, it is more like splitting all individual yearly heat flux cycles into cooling and warming segments and using those to “simulate” the thermosteric changes. For that part we refer to the seasonal cycles (now shown way ahead of everything) to partition the yearly data as just said (into cooling and warming segments). We admit that the present arrangement is not ideal, and doubt that there are universal recipes. However, we would appreciate receiving further reflections, recommendations and critiques.
--	---

Interpretations	
20	Reviewer #1: I will point out one issue in the interpretation of the results obtained in this study that I believe is problematic. We have confirmed a strong long-term correlation between SLA and SHA.
21	Reviewer #1: However, since SLA changes slightly earlier, I doubt the validity of the logic that SHA changes cause SLA changes. At the very least, a discussion should be added regarding the possible reasons why SLA appears to be leading.

	version of the long-term variability figure that included the seasonal cycles with one with the seasonal cycle subtracted, some seeming slight changes in timing become obvious. The reviewer probably refers to the slight shift of the tendency change that occurred around 1999 (although the ship-data of that year were insufficient for a precise timing of SHA), or to the low point between 2017-2018. We doubt any other shift is that obvious in our SLA and SHA. It is an interesting observation, for which we can offer, at very least, two possible explanations, which, unfortunately, cannot be added to the paper because we already exceeded the recommended word count (again, thanks for pointing out):  - For the reasons explained in our comment on the point in Row #20, the steric changes should really dominate the sea level change in the Labrador Sea as those are much larger in magnitude than the mass change equivalent. However, obeying to the third Newton's law, and similar principles of Mother Nature, the ocean may counteract by adding or removing physical mass aiming to compensate the stress. As a result, a SHA trend, like the one we see between 2013 and 2018, can be reversed in SLA slightly earlier as it rebounding from the drop; - There are interannual changes in the water column mass field, represented by the SLA-SHA differences in our figures, and if there is a trends in the mass that is added on top of SHA, the combined effect of the two can alter the phase of the resulting SLA. As a matter of fact, we see this clearly in 2017-2018. There had been a water column mass gain that occurred right at that point (could be a result of the Third-N-Law compensatory mechanism explained in the previous bullet, but we won't speculate about this is the paper), and just by superimposing it on the SHA minimum, we get an earlier termination of the negative trend shown to that point by SLA. On the other hand, the Reviewer might mean not as much the interannual timing, as the seasonal one. Indeed, Figure 3 shows a slight (~3 weeks?)
--	---

		lead of SLA in winter, but not in summer. This difference is reflected in the derived water column mass as a February low. The seasonal water column mass changes are driven by the seasonality of hydrological cycle and basin-wide circulation, like gyre dynamics. We discuss this in the closing section of the paper “Remaining challenges and future steps”. We are happy to see that our regard of oceanography as a fine high-accuracy trade, where attention to detail is a #1 rule, is shared by our Reviewers. It is just we sometimes stop at the point, where are present technological limitation do not allow us to make another step. This is probably where we are right now with the weak but distinct patterns shown by the residuals.
22	Reviewer #2: The paper does not make a strong case for the significance of central Labrador Sea sea level. I believe that this is a worthwhile topic because of the importance of the Labrador Sea for its role in AMOC and for the interesting currents and water mass formation there.	A wonderful statement! Defining the central Labrador Sea as a major receiving and transforming basin, a lung and a heart of the North Atlantic in our previous publications, we modestly underrepresented its significance this time. Even though we hinted at the three key aspects of sea level in the Introduction – (1) the impact on sea level as a whole, (2) the effects on the horizontal (through geostrophic current control) and vertical (through convergence/divergence) circulations, and hence (3) the role in water mass formation, we have reemphasized these and other aspects of the significance of sea level in both introduction final statements. In this revision, we have also strengthened the motivation for our study by emphasizing the importance of Labrador Sea Water formation (Lines 40–72). We now highlight that the central Labrador Sea (CLS) exhibits a rate of sea level rise that surpasses other deep basins in the subpolar North Atlantic (Lines 78–80). As the Reviewer insightfully suggested, we also provide broader context by comparing the CLS sea level to the global mean sea level rise in both the Introduction (Lines 80–88) and Discussion (Lines 502–513). This underscores that although halosteric contributions are minor in the global sea level budget, they exert a major influence on regional sea level variability in the Labrador Sea. These

		revisions are reflected in our concluding statements to better convey the broader climate significance of the CLS.
23	Reviewer #2: Paper references the importance of sea level for the [rather small] human populations along the Labrador and Greenland coasts, though the relationship is somewhat indirect given the complex spatial pattern of sea level rise [Figure 1]. This stuff seems very interesting to physical oceanography (my field) and somewhat interesting to anyone interested in sea level rise; not sure of how much interest it would be for others in climate.	Thank you for pointing this out. We have removed the original statement, and as noted in our previous response, we have rewritten the motivation for this study to better highlight the uniqueness of the central Labrador Sea. The revised text now focuses on its exceptional sea level behavior, physical setting, and role in subpolar dynamics, rather than coastal impacts.
24	Reviewer #2: Suggestions for Improvement, 2. Interpretation of results. Global sea level rise is a key feature of global warming. Paper needs to include more context from global sea level rise. For instance, Guerou et al (2023) show GMSL 10-year trends of about 2.5 cm/decade in 2000 and 4.4 cm/dec in 2015, which gives a reference point for what constitutes a “fast” sea level rise.	We thank the Reviewer for this important and constructive suggestion. In the revised manuscript, we have expanded both the Introduction and Discussion sections to better place the regional sea level rise in the central Labrador Sea (CLS) within the context of global sea level change. Specifically, we cite the range of short to long term global mean sea level trends reported in Figure 6a of Guerou et al. (2023), to provide a clearer benchmark for evaluating what constitutes a fast sea level rise. These comparisons help underscore the exceptional nature of the more than 10 cm sea level rise observed in the CLS between 2017 and 2025, which exceeds 1 cm per year. Additionally, we now emphasize in the Discussion that this regional acceleration is not only thermosteric in origin but also strongly halosteric, driven by changes in upper ocean salinity due to enhanced freshwater input from the Arctic. This halosteric amplification has received limited attention in previous studies, and we agree that it is important to highlight it here. These additions help clarify the broader significance of our findings in the context of global climate driven sea level changes.
25	Reviewer #2: Suggestions for Improvement, 2. Interpretation of results. To me it was very interesting how the	We appreciate the Reviewer’s observation and have expanded the Discussion section accordingly (Lines 502–513). We now elaborate on the

	SLR signal was dominated by steric height, given that globally, changes in steric height seems to account for less than half of the change in sea level (Dieng et al, 2017).	dominant steric contribution to sea level rise in the central Labrador Sea and contrast this with the global average, where steric changes account for about 30% ~40% of total sea level rise. In particular, we highlight the significant role of halosteric effects in driving regional trends—an aspect often underrepresented in earlier studies. These revisions aim to clarify the interpretation of our results in a broader context.
26	Reviewer #2: Suggestions for Improvement, 2. Interpretation of results. Other important factors include gravitational effects associated with geographical redistribution of mass, and changing circulation. I would think that there is some change in subpolar gyre strength which would change the sea surface height gradient between the edge and center of the gyre and hence change sea surface height in center of gyre.	We were actually thinking about looking at the gyre strength in relation to the water column mass change, but felt shy going too far in this direction, as it would take us in a different direction. To address this issue, we would need to compute or extract from the models the strength of barotropic circulation encompassing the central Labrador Sea, and related mass divergence. Instead, in discussion, we compare the summer and winter kinetic energy maps, showing wintertime intensification of the cyclonic gyre, i.e., increased mass divergence, agreeing with the sea level - steric height difference (Lines 530-542).
27	Reviewer #2: Suggestions for Improvement, 2. Interpretation of results. Finally, Labrador Sea is close to the “warming hole” of surface warming. When I look at 11 yr running mean (55-60 W, 55-60 N) from Berkeley Earth, I see a big jump in SST in the 1930s, a slow decline till around 1990, big jump over next 15 years, then slow decline again.	The reviewer made an excellent point! Honestly, when discussing long-term changes in the region, I try to stay away from two terms – the “warming hole” and “cold blob”. The first got stuck to the Labrador Sea because of some massive decadal scale cooling events in the 1970s, 1980s and 1990s (please also see Figure 8 in Intensification and shutdown of deep convection in the Labrador Sea were caused by changes in atmospheric and freshwater dynamics Communications Earth & Environment, which is entirely based on in-situ observations). These events tilted the regional centennial SST trend, whereas, as the reviewer correctly observed, there were some short-lived events (or artifacts in poorly-sampled area) associated with decadal shifts in the atmospheric dynamics, which can be partially attributed to NAO. Regardless of how we name it (we can still use the “warming hole” just to emphasize the uniqueness of the locations), the reviewer made an excellent point – since the climate change in the region is in contrasts we that outside, the sea level changes and

		trends are expected to be in contrast, and therefore a proper accurate and comprehensive analysis of sea level trend in the Labrador Sea is needed for better understanding of the complex dynamics of a larger region, and its evolution. Hopefully, we are on the same page with the reviewers in our understanding of this comment.
28	Reviewer #2: Seems like this is all worthy of discussion and it's possible that the dominance of steric effects is more surprising – hence interesting – than paper lets on.	We thank the Reviewer for this observation. As noted in our previous response, we have expanded the Discussion to emphasize the steric dominance in the central Labrador Sea and its contrast with the global sea level budget.
29	Reviewer #2: Suggestions for Improvement, 3. Significance of Record Highs. This point is closely related to #2. The paper highlights how the reversal of halosteric effects makes 2024 SLA slightly above previous records rather than slightly below. Does this small difference matter?	We thank the Reviewer for this thoughtful comment. We agree that the key issue is not the absolute difference in SLA, but the shift in the underlying mechanisms. As clarified in the revised Discussion (Lines 502–513), the record highs in 2023 to 2025 reflect a change in the steric balance, where halosteric effects began reinforcing rather than offsetting thermosteric expansion. This reversal helped push the SLA above previous records and signals a broader shift in how the Labrador Sea responds to freshwater forcing under ongoing climate change.
30	Reviewer #2: Suggestions for Improvement, 3. Significance of Record Highs. A bigger point, which paper does not say much about, is that variability on about 5 year timescales is strong enough to temporarily reverse the long-term upward trend.	This is an excellent point! We stopped short of explicitly explaining how the decadal sea level variations (of the order of 5-7 cm) in the Labrador Sea may alter, enhance or subdue the sea level trends observed in the rest of the Subpolar North Atlantic Ocean and the World Ocean as a whole. The reviewer is absolutely right here, a pentadal-to-decadal steric height cycle can easily topple the trend, like we saw in the early 1990s and between 2011 and 2018, but what we find particularly interesting here is that in the former case the effect of the cycle was pretty much neutral – the sea level recovered from the drop, catching up with the global trend, while in the recent cycle the sea level overcompensated the initial drop. This is a fundamental point, because if we earlier argued that decadal changes, mainly associated with convective cycles, dominate variability without leaving a noticeable residual trace on the trend, the recent steric cycle was asymmetrical raising sea

		level more than dropping, mainly because of the halosteric thermosteric relation change. On the other hand, and we added this to the text, because of the sustained intensification of winter convection during 2012-2017, the resulting drop of sea level fully compensated the rise that occurred over the previous/preceding twenty years, reducing the 1998-2017 sea level change to negative. Once again, we thank the reviewer for encouraging us to expand this point in the manuscript.
Details		
31	Reviewer #2: Some Details, Fig 1  * Where does velocity and sea level data come from? * Why show trends in highs and lows (bottom panels)? * Showing the trends makes me want paper to say more about them. It looks like the surface geostrophic current around the basin is not getting stronger, though the Lab Current is. It's a little distracting, since rest of paper focuses on CLS temporal behavior. 	We thank the Reviewer for these constructive observations. In the revised caption to Figure 1, we have clarified that the velocity fields represent surface geostrophic currents derived from satellite altimetry. In addition, we now clearly state in the Data Sources section that gridded altimetry products were used to construct Figures 1 and 9 (Lines 664–667). Our choice to use gridded altimetry is twofold: (1) it provides sufficient spatial accuracy to highlight the uniqueness of the CLS, and (2) it significantly reduces the data-processing workload without compromising scientific integrity. In the previous version of Figure 1, we included trend maps of annual sea level lows and highs to emphasize the seasonality of freshwater effects, particularly during summer. However, we realized that this approach diluted the core message of the figure: the distinctiveness of the CLS. In the revised figure, we instead place the CLS in the context of the broader Subpolar North Atlantic and compare its signal against the global mean sea level rise, as the Reviewer helpfully suggested. To avoid confusion and sharpen focus, we now show only the recent trends over the period 2017–2025, which better illustrates the accelerated regional changes. We also emphasize in both the caption and the main text that these spatial trends are intended to provide broader context, while our

		central focus remains on the temporal evolution within the CLS (Lines 78–88).
32	Reviewer #2: Some Details Fig 2 and some subsequent figures. Make separate figures to show annual cycle and interannual variability. For annual cycle and monthly (weekly?) data points, select a subset (10 years?) and superimpose fitted curves shown in current version of figure. In separate figure, show low-frequency data as before, but with altimetry and 10-3300 dbar hydrography superimposed and either no annual cycle or annual cycle replaced by one curve connecting all the annual maxima of fitted curve and another curve connecting annual mins. Most of the discussion of these figures is about interannual variability and comparisons between different quantities (height, thermocline component, etc) for which the seasonal cycle is a distraction.	This is an excellent suggestion. As we explained before, we removed the seasonal cycles from all multiyear variability figures, and moved the seasonal cycle figures to the start of the “Results” (Figures 3-4 and Supplementary Figures 5-6). All time series in the main manuscript are de-seasoned as the Reviewer rightly suggested. On the other hand, to those who want to see how the full variability cycles look like, we added the figures with the regular seasonal cycles added to the time series to the Supplementary (Supplementary Figure 7-8).
33	Reviewer #2: Some Details Lines 139-176 & Fig 4. Very hard to read. First sentence is painful. It is followed by description of Labrador Sea budgets which maybe should go in Intro when motivating the analysis of changes in steric heights. It’s hard to see the breaks in deep convection from Fig 4. If deep convection is signaled by deep annual maximum in bottom of mixed layer, then temperature, salinity, and density fields all look to me like they are showing different patterns. It’s also hard to detect these breaks given aspect ratio of figure. I think the bottom panel is showing how deep one must go for density to be 0.005 kg/m³ denser than surface, but I’m not sure. It’s confusing that shut-downs of deep convection are listed in first paragraph, but then how one monitors deep convection from property fields is	We thank the Reviewer for raising this concern as we also found that part hard to read and felt ashamed of the poorly written text concerning the key mechanisms of interannual changes. The whole text has been rewritten, and part of it moved to the “Key Research Questions” section. Furthermore, we did clearly explain in the previous version why temperature, salinity, density and pycnostad look so different in previous Figure 4 (present Figure 2). Even though the earlier versions of this figure have been discussed in our previous publications in detail (references added), we agree with the Reviewer that need to explain, at least briefly, why temperature and salinity show different patterns. This has been done, and we also explain the meaning of the bottom panel of present Figure 2 – the purpose of this diagram is not as much to show the density of the surface (it can vary from location to location, and, therefore, we never use the “traditional” density threshold method, e.g., density within 0.005 kg/m³), but to identify thick uniform density layers (pycnocline)

	described in the middle of second paragraph.	that we use as a reference point for convectively-formed water, mixed layer depth, etc. This is how we identify and map the product of winter convection.
34	Reviewer #2: Some Details Equations 1-3. Simplify notation such as removing subscripts from S_A, V_sp, T_C and using subscript m rather than mean to denote mean. Typo line 66 “equations xx”.	We have revised the notation in Equations 1–3 for improved clarity and consistency. Specifically, we removed unnecessary subscripts and now use the subscript m to denote mean values (lines 600–612). We also corrected the typo on line 66 of the previous manuscript version by replacing “equations xx” with the correct equation reference.
35	Reviewer #2: Some Details Lots of typos: Please do some additional proofreading. Many instances of “heigh” instead of “height”, subsection title “Where there other occurrences...” instead of “Were there other occurrences...”	We thank the Reviewer for pointing out these issues. We have carefully proofread the manuscript and corrected all identified typographical errors, including replacing “heigh” with “height” and correcting the subsection title to read “Were there other occurrences...” as intended. We also performed a full spell-check and line-by-line review to ensure overall consistency and clarity throughout the manuscript.
36	Reviewer #2: Some Details Long subsection titles. Paper subsection titles make actual statement of fact, which is useful, but long section titles are harder to read.	We thank the Reviewer for bringing this up. The subsection titles have been shortened and edited as recommended, as shown in Lines of 209, 240, 260, 333, 404, and 470 in the revised manuscript.
37	Reviewer #2: Some Details Winter convection regime changes as the primary driver of sea level variability in the Labrador Sea  Winter convection regime changes cause sea level variability	Thanks. We have revised this subsection title (Line 240) to “Winter convection regime changes drive sea level variability” for improved clarity and directness.
38	Reviewer #2: Some Details A reversal in the upper layer temperature-salinity steric balance drove the sea level to a record high  Reversal of temperature-salinity relation drove record high sea level	Thanks. We have revised this subsection title (Line 260) as “Reversal of the upper layer’s temperature-salinity relationship accelerated the sea level rise” to improve clarity and accurately reflect the underlying mechanism.

39	Reviewer #2: Some Details Were there other occurrences of halosteric reinforcements of thermosteric heigh changes in the Labrador Sea in the past?  Pre-1990 halosteric reinforcements of thermosteric height changes	We appreciate the Reviewer’s suggestion to simplify the phrasing. However, we have chosen to retain the question-style formulation: “Has a halosteric reinforcements of a thermosteric trend ever occurred in the past?” This version aligns better with the surrounding context and clarifies that the question refers to broader temporal behavior, not just a specific pre-1990 interval.
-----------	---	---

Sincerely,
 Igor Yashayaev and Yang Zhang

Response to the Reviewers' Comments on

Concurrent warming, freshening and cessation of deep convection in the Labrador Sea raised its sea level to a record high

Igor Yashayaev¹, Yang Zhang^{2,3}

¹Bedford Institute of Oceanography, Dartmouth, N.S., Canada

²School of Marine Science and Policy, University of Delaware, Lewes, Delaware, USA

³Scripps Institution of Oceanography, University of California, San Diego, La Jolla, CA, USA

Corresponding author: Igor Yashayaev

Emails: Igor.Yashayaev@dfo-mpo.gc.ca; Labrador.Sea@gmail.com

Phone: (782)-640-9437

Reviewer #1 (Remarks to the Author):

The authors responded sincerely to the reviewers' comments. I have confirmed in the Discussion that, regarding my concern about the timing mismatch in the temporal variations of SHA and SLA, it is not easy to demonstrate this with the current observational data, and that, in addition to changes in basin-scale circulation, other possible causes for this mismatch are also mentioned. I believe that the manuscript is suitable for publication as a paper in Nature Communications.

Response:

The authors are thankful to the Reviewer for attentive detailed constructive review. We totally agree with the reviewer about the present data limitations in handling second-order difference. However, we mentioned a few factors in the present revision which may explain slightly mismatched timing in the respective interannual signals and seasonal cycles.

Reviewer #2 (Remarks to the Author):

This is a review of the revised manuscript. The revision is better organized and clearer. It tells an interesting story of rapid sea level rise in the Labrador Sea driven by thermal expansion and augmented by a recent change in salinity effects, both of which the authors related to atmospheric forcing. I recommend publication.

Response:

The authors highly appreciate such a thoughtful point. Indeed, even though we attribute the halosteric height rise to the Arctic sea ice cover reduction and increase inflow meltwater, the undeniable fact behind this change is Arctic Amplification driven by atmospheric forcing. This is a great point about the fundamental cause of the revealed sea level changes.

I have a few remaining suggestions which I leave to the authors' discretion; I do not need to review the manuscript again.

1) Correspondence between numbered questions in Key Research Questions and subsections in Results. The Results subsections partly correspond to the questions. Perhaps it would be helpful for readers if each question corresponding to a results section. Questions 3-5 all pertain to halosteric effects and could be combined into a single item. The last couple of sections raise and answer questions that are not mentioned in the original listing.

Response:

The authors are thankful for this great suggestion. Indeed, there are more questions answered than raised. Two questions have been added answered one-by-one in two remaining subsections of the results. To keep the question-to subsection correspondence throughout the work, also noting the attention raised to the halosteric component is novel and questions 3-5 logically follow one from the other, they are still kept apart. However, all the questions have been carefully rewritten adding to their uniqueness.

2) Figure 1ab. For filled contour maps, in general I recommend using a small number of discrete shades/colors, rather than a continuous or near-continuous color palette. This makes it easier to see the range of values present in a field. Currently it is hard to say much more than that there is a lower (green) and higher (purple) part of the gyre in 1a and a rapidly (purple) and less rapidly (white) rising sea surface in 1b. If the authors want to show any gradations beyond that, a few discrete shades would be helpful.

Response:

Paying much attention to the visual aspects of the paper, the authors feel ashamed to admit it was planned by them to critically revise and update all figure, starting with Figure, which needed more than enhanced color gradation. It needed labels for the Labrador and West-Greenland Currents, indication of the global trend, and entire realignment of all graphical details. Given that it is the first figure, it needed to be improved, and the Editor's comment made it clear. The figure has been fully redesigned. Instead of discrete colors, a continuous with clearer color gradation and transitions is used. One of the reasons not to switch to discrete was staying consistent through paper and Supplementary Figures.